# Heri-Graphs: A Dataset Creation Framework for Multi-Modal Machine Learning on Graphs of Heritage Values and Attributes with Social Media

Nan Bai [1,*], Pirouz Nourian [2], Renqian Luo [3] and Ana Pereira Roders [1]

1   UNESCO Chair in Heritage and Values: Heritage and the Reshaping of Urban Conservation for Sustainability, Delft University of Technology, 2628 BL Delft, The Netherlands
2   Genesis Lab of Generative Design and Generative Science, Delft University of Technology, 2628 BL Delft, The Netherlands
3   Microsoft Research, Beijing 100080, China
*   Correspondence: n.bai@tudelft.nl

**Abstract:** Values (why to conserve) and Attributes (what to conserve) are essential concepts of cultural heritage. Recent studies have been using social media to map values and attributes conveyed by the public to cultural heritage. However, it is rare to connect heterogeneous modalities of images, texts, geo-locations, timestamps, and social network structures to mine the semantic and structural characteristics therein. This study presents a methodological framework for constructing such multi-modal datasets using posts and images on Flickr for graph-based machine learning (ML) tasks concerning heritage values and attributes. After data pre-processing using pre-trained ML models, the multi-modal information of visual contents and textual semantics are modelled as node features and labels, while their social relationships and spatio-temporal contexts are modelled as links in Multi-Graphs. The framework is tested in three cities containing UNESCO World Heritage properties—Amsterdam, Suzhou, and Venice— which yielded datasets with high consistency for semi-supervised learning tasks. The entire process is formally described with mathematical notations, ready to be applied in provisional tasks both as ML problems with technical relevance and as urban/heritage study questions with societal interests. This study could also benefit the understanding and mapping of heritage values and attributes for future research in global cases, aiming at inclusive heritage management practices. Moreover, the proposed framework could be summarized as creating attributed graphs from unstructured social media data sources, ready to be applied in a wide range of use cases.

**Keywords:** World Heritage; Flickr; multi-modal dataset; graph construction; machine and deep learning

## 1. Introduction

In the context of the UNESCO World Heritage (WH) Convention, "values" (why to conserve) and "attributes" (what to conserve) have been used extensively to detail the cultural significance of heritage [1,2]. Meanwhile, researchers have provided categories and taxonomies for heritage values and attributes, respectively, [3–5]. Both concepts are essential for understanding the significance and meaning of cultural and natural heritage, and for formulating more comprehensive management plans [5]. However, the heritage values and attributes are not only to define the significance of Outstanding Universal Value (OUV) in the specific context of the World Heritage List (WHL), but all kinds of significance, ranging from listed to unlisted, natural to cultural, tangible to intangible, and from global to national, regional and local [4,6–9]. Moreover, the 2011 UNESCO *Recommendation on the Historic Urban Landscape* (HUL) stressed that heritage should also be recognized through the lens of local citizens, tourists and experts, calling for tools for civic engagement and knowledge documentation [8–10].

Thereafter, in the past decade, analyses have been performed on User-Generated Content (UGC) from social media platforms to actively collect opinions of the [online] public, and to map heritage values and attributes conveyed by various stakeholders in urban environments [11,12]. In Machine Learning (ML) literature, a *modality* is defined as "*the way in which something happens or is experienced*", which can include natural language, visual contents, vocal signals, etc., [13]. Most of the studies mapping heritage values and attributes from UGC focused only on a few isolated modalities, such as textual topics of comments and/or tags [14–16], visual content of depicted scenes [17,18], social interactions [19–21], and geographical distribution of the posts [22,23].

However, the heterogeneous multi-modal information from social media can enrich the understanding of posts, as textual and visual content, temporal and geographical contexts, and underlined social network structures could show both complementary and contradictory messages [9,24]. A few studies have analysed different modalities to reveal the discussed topics and depicted scenes about cultural heritage [25,26]. However, since they (mostly) adapted analogue approaches during analyses and the multi-modal information was not explicitly paired, linked, and analysed together, these studies could not yet be classified as *Multi-modal Machine Learning* (MML), aiming to "*build models that can process and relate information from multiple modalities*" [13] to enrich the conclusions that could not be easily achieved with isolated modalities. On the other hand, Crandall et al. [27] proposed a global dataset collected from Flickr with visual and textual features, as well as geographical locations. Graphs were constructed with multi-modal information to map, cluster, and retrieve the most representative landmark images for major global cities. Gomez et al. [28] trained multi-modal representation models of images, captions, and neighbourhoods with Instagram data from within Barcelona, able to retrieve the most relevant photos and topics for each municipal district, being used to interpret the urban characteristics of different neighbourhoods. More recently, the continuous research line demonstrated in Kang et al. [29] and Cho et al. [30] applied transfer learning [31] techniques to classify geo-tagged images into hierarchical scene categories and connected the depicted tourist activities to the urban environments where these cultural activities took place. Although not all of them explicitly referred to heritage, these studies could provide useful information for scholars and practitioners to gather knowledge from the public about their perceived heritage values and attributes in urban settings, as suggested by HUL [9,10]. Among the five main MML challenges summarized by Baltrusaitis et al. [13], representation (*to present and summarize multi-modal data in a joint or coordinated space*) and fusion (*to join information for prediction*) can be the most relevant ones for heritage and urban studies, with respect to (1) retrieving visual and/or textual information related to certain heritage values and attributes, and (2) aggregating individual posts in different geographic and administrative levels as the collective summarized knowledge of a place.

Furthermore, according to the *First Law of Geography* [32], "*everything is related to everything else, but near things are more related than distant things*". This argument can also be assumed to be valid in distance measures other than geographical ones where a random walk could be performed [33], such as in a topological space abstracted from spatial structure [34–37] or a social network constructed based on common interests [38–41]. In this light, it would be beneficial to construct graphs of UGC posts where Social Network Analysis (SNA) could be performed, showing the socio-economic and spatio-temporal context among them, reflecting the inter-related dependent nature of the posts [42]. Such a problem definition could help with both the classification and the aggregation tasks mentioned above, as has been demonstrated as effective and powerful by applications in the emerging field of Machine and Deep Learning on Graphs [43,44].

This paper describes the methodological framework of creating multi-modal graph-based datasets about heritage values and attributes using unstructured social media data. The core question from generating such datasets could be formulated as: while heritage values and attributes have been historically inspected from site visiting and document reviewing by experts, can computational methods and/or artificial intelligence aid the

process of knowledge documentation and comparative studies by mapping and mining multi-modal social media data? Even if acceleration of the processes is not a priority, the provision of such a framework is aimed to encourage consistency and inclusion of communities in the discourse of cherishing, protecting, and preserving cultural heritage. In other words, the machine can eventually represent the voice of the community [9]. The main contributions of this manuscript can be summarized as:

1. Domain-specific multi-modal attributed graph datasets about heritage values and attributes (or more precisely, the values and attributes conveyed by the public to urban cultural heritage) are collected and structured with user-generated content from the social media platform Flickr in three cities (Amsterdam, Suzhou, and Venice) containing UNESCO World Heritage properties, which could benefit the knowledge documentation and mapping for heritage and urban studies, aiming at a more inclusive heritage management process;

2. Several pre-trained machine learning and deep learning models have been extensively applied and tested for generating multi-modal features and [pseudo-]labels with full mathematical formulations as its problem definition, providing a reproducible methodological framework that could also be tested in other cases worldwide;

3. Multi-graphs have been constructed to reflect the temporal, spatial, and social relationships among the data samples of collected user-generated content, ready to be further tested on several provisional tasks with both scientific relevance for Graph-based Multi-modal Machine Learning and Social Network research, and societal interests for Urban Studies, Urban Data Science, and Heritage Studies.

## 2. Materials and Methods

### 2.1. General Framework

Before delving into the domain-specific case studies with technological details, this section first describes the general process of creating multi-modal datasets as attributed graphs from unstructured volunteered information content harvested from social media. These graphs would encode connections between posts of content publishers on social media; connections that can be established by virtue of similarities or proxmities in spatial, temporal, or social domains. The whole process consists of five core components: data acquisition and cleaning (Section 2.3), multi-modal feature representation (Section 2.4), [pseudo-] label generation (Section 2.5), contextual graph construction (Section 2.6), and qualitative inspection and validation (Section 3), as visualized in Figure 1.

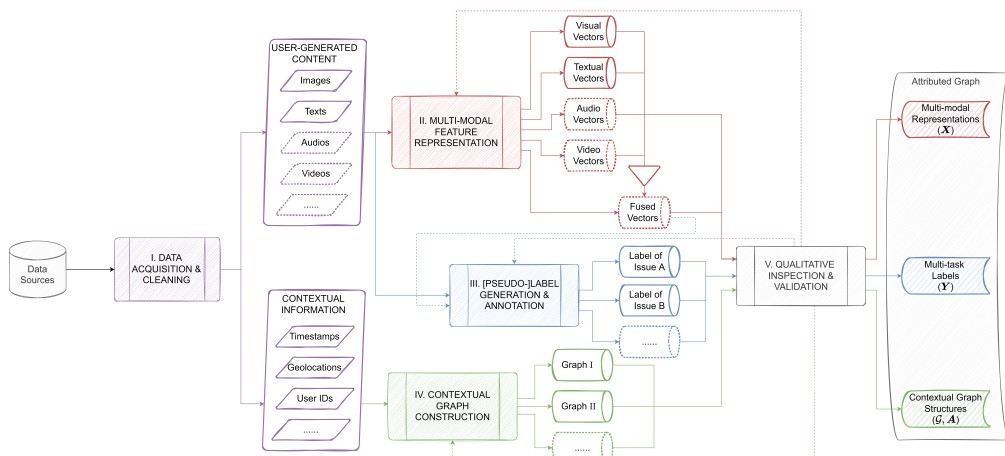

**Figure 1.** The framework to create multi-modal machine learning datasets as attributed graphs from unstructured data sources.

As argued in Aggarwal [24] and Bai et al. [9], the analyses on social media (or social network data) could be categorized as studying its content (traditionally texts and images,

possibly also audio and video), structure (social linkages among users entailing interactions), and context (spatio-temporal and socio-economic manifolds). While the former is mainly about constituent data points themselves, the latter two (both are contextual information under different scenarios) provide explicit data about the potential linkage between the data points. For any data source (social media platform) of interest, the proposed framework suggests acquiring both content and contextual information for a rigid understanding of the social network. After data acquisition and cleaning, the input data would be highly unstructured and non-standard and thus challenging to feed into data science workflows, which need to be transformed as machine-readable formats-presumably vectors-using certain feature representation techniques. For different modalities, various techniques could be employed: from hand-engineered features, to pre-trained embeddings, and to other end-to-end techniques such as auto-encoders. Moreover, the fusion of different modalities could happen in various forms, from the most simple concatenation, to more complex techniques using neural networks [13]. Even though unsupervised learning applications of spatial clustering and auto-correlation are not uncommon, it is still preferable to have semantic labels concerning various issues of interest to make more sense out of the data points. In situations where human annotation can be expensive and challenging, semi-automatic labeling with transfer learning, pseudo-label generation and/or active learning using either the raw data or the generated multi-modal features could be applied to efficiently circumvent this bottle-neck process [31,45–49]. Furthermore, the graph construction process makes use of the proximity or similarity of the contextual information to link the data points as [multi-] graphs. Contextualization of the data points and creating a coherent picture of the datasets are necessary tasks, without which the task of data analysis would remain at the level of dealing with a bag containing powder-like data points. Graph datasets can be of essential value in interpolation and extrapolation tasks, simply put for diffusing or transferring information from the neighbours of a data point to it. In cases where some graph attribute is missing on a data point, a graph representation can help in creating consistency and coherence. This is especially important for semi-supervised learning scenarios on social media data, where missing features could be very common [50]. Before storing the results as valid attributed graph datasets with graph structures ($\mathcal{G}$, $A$) and node features ($X$, $Y$), a bundle of processes for qualitatively and quantitatively inspecting the quality, consistency, and validity of the generated results is necessary. This could also possibly include humans in the loop.

The rest of this manuscript will explain each component in detail with specific instances tailored for the use case of mapping heritage values and attributes as demonstration (such as the selection of the three cities in Section 2.2, the choice of Flickr as data source in Section 2.3, and the decisions of pre-trained ML models in Sections 2.4 and 2.5). However, in principle, the case study to be instantiated and technology to be employed could be specified, enhanced, and updated based on the actual use cases within a wide range of scenarios, taking advantage of the most suitable tools and the most current technological developments. This will be further discussed in Section 4.2.

### 2.2. Selection of Case Studies

Without loss of generality, this research selected three cities in Europe and China that are related to UNESCO WH and HUL as case studies: Amsterdam (AMS), the Netherlands; Suzhou (SUZ), China; and Venice (VEN), Italy. All three cities themselves are either entirely or partially inscribed in the WHL, such as *Venice and its Lagoon* (https://whc.unesco.org/en/list/394, accessed on 8 March 2022) and *Seventeenth-Century Canal Ring Area of Amsterdam inside the Singelgracht* (https://whc.unesco.org/en/list/1349/, accessed on 8 March 2022), or contain WHL in multiple parts of the city, such as the *Classical Gardens of Suzhou* (http://whc.unesco.org/en/list/813, accessed on 8 March 2022), showcasing different spatial typologies of cultural heritage in relation to its urban context [51,52].

As shown in Table 1, the three cases have very different scales, yet all strongly demonstrate the relationship between urban fabric and the water system. Interestingly, Amsterdam

and Suzhou have been, respectively, referred to as "*the Venice of the North/East*" by the media and public. Moreover, the concept of OUV introduced in Section 1 reveals the core cultural significance of WH properties. The OUV of a property would be justified with ten selection criteria, where criteria (i)–(vi) reflect various cultural values, and criteria (vii)–(x) natural ones [2,53–55], as explained in Appendix Table A3. The three selected cases include a broad variety of all cultural heritage OUV selection criteria, implying the representativeness of the datasets constructed in this study.

**Table 1.** The case studies and their World Heritage status.

| City | Geo-Location | WHL Name | OUV Criteria | Area of Property | Inscription Date |
|---|---|---|---|---|---|
| Amsterdam (AMS) | 52.365000N 4.887778E | *Seventeenth-Century Canal Ring Area of Amsterdam inside the Singelgracht* | (i), (ii), (iv) | 198.2 ha | 2010 |
| Suzhou (SUZ) | 31.302300N 120.631300E | *Classical Gardens of Suzhou* | (i), (ii), (iii), (iv), (v) | 11.9 ha | 2000 |
| Venice (VEN) | 45.438759N 12.327145E | *Venice and its Lagoon* | (i), (ii), (iii), (iv), (v), (vi) | 70,176.2 ha | 1987 |

*2.3. Data Collection and Pre-Processing*

Numerous studies have collected, annotated, and distributed open-source datasets from the image-sharing social media platform Flickr owing to its high-quality image data, searchable metadata, and convenient Application Programming Interface (API), although its possible drawbacks include relatively low popularity, limited social and geographical coverage of users, and unbalanced information quantities of images and texts [56–59]. A collection of Flickr-based datasets could include *MirFlickr-1M* [60], *NUS-WIDE* [61], *Flickr* [62], *ImageNet* [63,64], Microsoft Common Object in COntext (*MS COCO*) [57], *Flickr30k* [65], *SinoGrids* [66], and *GRAPH Saint* [67], etc. These datasets containing one or more of the visual, semantic, social, and/or geographical information of UGC are widely used, tested, but also sometimes challenged by different ML communities including Computer Vision, Multi-modal Machine Learning, and Machine Learning on Graphs. However, they are more or less suitable for bench-marking general ML tasks and testing computational algorithms, which are not necessarily tailor-made for heritage and urban studies. On the other hand, the motivation of data collection in this research is to provide datasets that could be both directly applicable for ML communities as a test-bed, and theoretically informative for heritage and urban scholars to draw conclusions on for informing the decision-making process. Therefore, instead of adapting the existing datasets that can be weakly related to the problems of interest in this study, new data are directly collected and processed from Flickr as an instance of the proposed framework in Section 2.1. Further possibilities of merging other existing datasets and data from other sources in response to the limitations of Flickr will be briefly addressed in Section 4.2.

FlickrAPI python library (https://stuvel.eu/software/flickrapi/, accessed on 8 March 2022) was used to access the API method provided by Flickr (https://www.flickr.com/services/api/, accessed on 8 March 2022), using the Geo-locations in Table 1 as the centroids to search a maximum of 5000 IDs of geo-tagged images within a fixed radius covering the major urban area, to form comparable and compatible datasets from the three cities, since only 4229 IDs were found in Suzhou during the time of data collection, reflecting the relatively scarse use of Flickr in China. To test the scalability of the methodological workflow, another larger dataset without an ID number limit has also been collected in Venice (VEN-XL). Only images with a `candownload` flag indicated by the owner were further queried, respecting the privacy and copyrights of Flickr users. The following information of each post was collected: owner's ID; owner's registered location on Flickr; the title, description, and tags provided by user; geo-tag of the image; timestamp marking when the image was taken, and URLs to download the `Large Square` (150 × 150 px) and `Small 320` (320 × 240 px) versions of the original image. Furthermore, the public friend and subscription lists of all the retrieved owners were queried, while all personal information was only considered as a [semi-] anonymous ID with respect to the privacy policy. The data

collection procedure took place from 28 December 2020–10 January 2021 and 10 February 2022–25 February 2022, respectively. The earliest captured photos collected date back to 1946 in AMS, 2007 in SUZ, 1954 in VEN, and 1875 in VEN-XL, and for all cities the most recent photos were taken in 2021–2022.

The retrieved textual fields of `description`, `title`, and `tags` could all provide useful information, yet not all posts have these fields, and not all posts are necessarily written to express thoughts and share knowledge about the place (considered as "valid" in the context of this study). The textual fields of the posts were cleaned, translated, and merged into a `Revised Text` field as raw English textual data, after recording the detected original language of posts on the sentence level using Google Translator API from the Deep Translator Python library (https://deep-translator.readthedocs.io/en/latest/, accessed on 8 March 2022). Moreover, many posts shared by the same user were uploaded at once, thus having the same duplicated textual fields for all of them. To handle such redundancy, a separate dataset of all the unique processed textual data on sentence level was saved for each city, while the original post ID of each sentence was marked and could easily be traced back.

Detailed description of the data collection and pre-processing procedure could be found in Appendix A. Table 2 shows the number of data samples (posts) and owners (users) for the three case study cities at each stage. Note the numbers of posting owners are relatively unbalanced in different cities. Intuitively, a larger number of owners could suggest a better coverage of social groups and provide better representativeness for the datasets. However, since the unit of data points in this study is a single post, not a unique social media user (content publisher), it could be assumed that the latter only provides sufficient [social] contextual information for the former.

**Table 2.** The number of data samples collected at each stage, the bold numbers mark the sample size of the final datasets.

| City | AMS | SUZ | VEN | VEN-XL |
|---|---|---|---|---|
| IDs Collected | 5000 | 4229 | 5000 | 116,675 |
| Is Downloadable | 3727 | 3137 | 2952 | 80,964 |
| **Downloaded Posts** | **3727** | **3137** | **2951** | **80,963** |
| Has Textual Data * | 3404 | 2692 | 2801 | 77,644 |
| Has Unique Texts ** | 3130 | 1963 | 1952 | 59,396 |
| Unique Sentences | 2247 | 361 | 3249 | 61,253 |
| Original Posts ** | 2904 | 754 | 1761 | 49,823 |
| Posting Owners | 195 | 95 | 330 | 6077 |

\* At least one of `Description`, `Title` and `Tag` fields is not empty. ** The two rows of numbers are different because of posts without any *valid* sentences.

To formally describe the data, define the problem, and propose a generalizable workflow as a methodological framework, mathematical notations are used in the rest of this manuscript. Since the same process is valid for all three cities (and probably also for other unselected cases worldwide) and has been repeated exactly three times, no distinctions would be made among the cities, except for the cardinality of sets reflecting sample sizes. Let $i$ be the index of a generic sample of the dataset for one city, then its raw data could be denoted as a tuple $\mathfrak{d}_i = (\mathfrak{I}_i, \mathcal{S}_i, \mathfrak{u}_i, \mathfrak{t}_i, \mathfrak{l}_i), \mathfrak{d}_i \in \mathfrak{D} = \{\mathfrak{d}_1, \mathfrak{d}_2, \ldots, \mathfrak{d}_K\}$, where $K$ is the sample size of the dataset in a city (as shown in Table 2), $\mathfrak{I}_i$ is a three-dimensional tensor of the image size with three RGB channels, $\mathcal{S}_i = \{\int_i^{(1)}, \int_i^{(2)}, \ldots, \int_i^{(|\mathcal{S}_i|)}\}$ or $\mathcal{S}_i = \varnothing$ is a set of revised English sentences that can also be an empty set for samples without any valid textual data, $\mathfrak{u}_i \in \mathcal{U}$ is a user ID that is one instance from the user set $\mathcal{U} = \{\mu_1, \mu_2, \ldots, \mu_{|\mathcal{U}|}\}$, $\mathfrak{t}_i \in \mathcal{T}$ is a timestamp that is one instance from the ordered set of all the unique timestamps $\mathcal{T} = \{\tau_1, \tau_2, \ldots, \tau_{|\mathcal{T}|}\}$ from the dataset at the level of weeks, and $\mathfrak{l}_i = (\mathfrak{x}_i, \mathfrak{y}_i)$ is a geographical coordinate of latitude ($\mathfrak{y}_i$) and longitude ($\mathfrak{x}_i$) marking the geo-location of the post. A complete nomenclature of all notations used in this paper can be found in the Appendix

Tables A1 and A2. Figure 2 demonstrates the data flow of one sample post in Venice, which will be formally explained in the following sections.

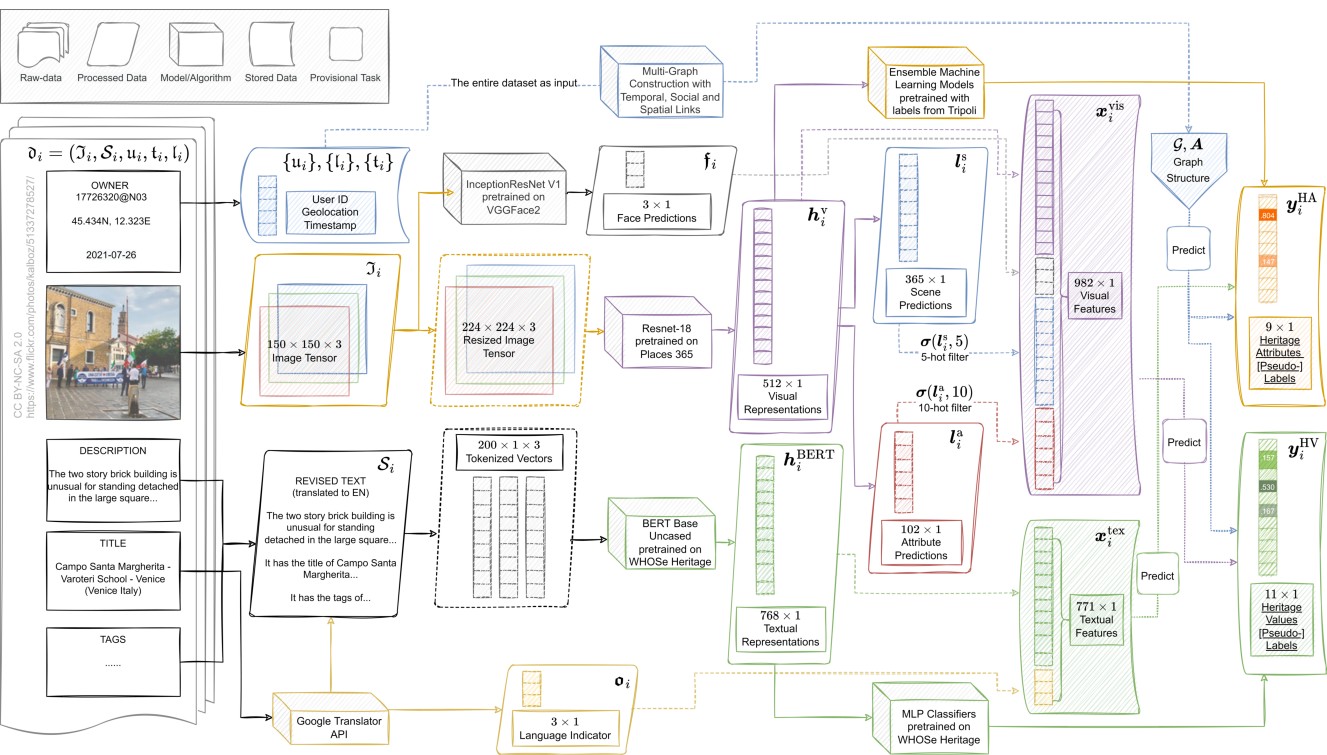

**Figure 2.** Data flow of the multi-modal feature generation process of one sample post in Venice, while graph construction requires all data points of the dataset. The original post owned by user `17726320@N03` is under CC BY-NC-SA 2.0 license.

### 2.4. Multi-Modal Feature Generation

#### 2.4.1. Visual Features

*Places365* is a dataset containing 1.8 million images from 365 scene categories, which includes a relatively comprehensive collection of indoor and outdoor places [68,69]. The categories can be informative for urban and heritage studies to identify depicted scenes of images and to further infer heritage attributes [5,26]. A few Convolutional Neural Network (CNN) models were pre-trained by Zhou et al. [69] using state-of-the-art backbones to predict the depicted scenes in images, reaching a top-1 accuracy of around 55% and top-5 accuracy of around 85%. Furthermore, the same set of pre-trained models have been used to predict 102 discriminative scene attributes based on *SUN Attribute* dataset [70,71], reaching top-1 accuracy of around 92% [69]. These scene attributes are conceptually different from heritage attributes, as the former are mostly adjectives and present participles describing the scene and activities taking place. Therefore, both heritage values and attributes could be effectively inferred therefrom.

This study used the openly-released ResNet-18 model [72] pre-trained on *Places365* with PyTorch (https://github.com/CSAILVision/places365, accessed on 9 March 2022). This model was adjusted to effectively yield three output vectors: (1) the last softmax layer of the model $l^{\mathrm{s}}_{365\times1}$ as logits (formally a probability distribution) over all scene categories; (2) the last hidden layer $h^{\mathrm{v}}_{512\times1}$ of the model; (3) a vector $l^{\mathrm{a}}_{102\times1}$ as logits over all scene attributes. Such a process for any image input $\mathfrak{I}_i$ could be described as:

$$l^{\mathrm{s}}_i, l^{\mathrm{a}}_i, h^{\mathrm{v}}_i = f_{\mathrm{ResNet\text{-}18}}(\mathfrak{I}_i | \Theta_{\mathrm{ResNet\text{-}18}}), \qquad (1)$$

or preferably in a vectorized format:

$$\boldsymbol{L}^{\mathrm{s}}, \boldsymbol{L}^{\mathrm{a}}, \boldsymbol{H}^{\mathrm{v}} = f_{\mathrm{ResNet\text{-}18}}([\mathfrak{I}_1, \mathfrak{I}_2, \ldots, \mathfrak{I}_K] | \Theta_{\mathrm{ResNet\text{-}18}}), \tag{2}$$

where

$$\boldsymbol{L}^{\mathrm{s}} := [l_i^{\mathrm{s}}]_{365 \times K}, \boldsymbol{L}^{\mathrm{a}} := [l_i^{\mathrm{a}}]_{102 \times K}, \boldsymbol{H}^{\mathrm{v}} := [h_i^{\mathrm{v}}]_{512 \times K}. \tag{3}$$

Considering that the models demonstrate reasonable performance in top-$n$ accuracy, to keep the visual features explainable, a $n$-hot soft activation filter $\sigma^{(n)}$ is performed on both logit outputs, to keep the top-$n$ prediction entries active, while smoothing all the others based on the confidence of top-$n$ predictions ($n = 5$ for scene categories $\boldsymbol{L}^{\mathrm{s}}$ and $n = 10$ for scene attributes $\boldsymbol{L}^{\mathrm{a}}$). Let $\max(\boldsymbol{l}, n)$ denote the $n_{\mathrm{th}}$ maximum element of a $d$-dimensional logit vector $\boldsymbol{l}$ (the sum of all $d$ entries of $\boldsymbol{l}$ equals 1), then the activation filter $\sigma^{(n)}$ could be described as:

$$\sigma^{(n)}(\boldsymbol{l}_{d \times 1}) = \boldsymbol{l} \odot \boldsymbol{m} + \frac{1 - \boldsymbol{l}^{\mathsf{T}} \boldsymbol{m}}{d - n}(\boldsymbol{1}_{d \times 1} - \boldsymbol{m}), \tag{4}$$

$$\boldsymbol{m} := [m_\iota]_{d \times 1}, m_\iota = \begin{cases} 1 & \text{if } l_\iota \geq \max(\boldsymbol{l}, n) \\ 0 & \text{otherwise} \end{cases}, \tag{5}$$

where $\boldsymbol{m}$ is a mask vector indicating the positions of top-$n$ entries, and $\boldsymbol{l}^{\mathsf{T}} \boldsymbol{m}$ is effectively the total confidence of the model for top-$n$ predictions. Note that this function could also take a matrix as input and process it as several column vectors to be concatenated back.

Furthermore, as the *Places365* dataset is tailor-made for scene detection tasks rather than facial recognition [69], the models pre-trained on it may become confused when a new image is mainly composed of faces as "typical tourism pictures" and self-taken photos, which is not uncommon in the case studies as popular tourism destinations. As the ultimate aim of constructing such datasets is not to precisely predict the scene each image depicts, but to help infer heritage values and attributes, it would be unfair to simply exclude those images containing a significant proportion of faces. Rather, the existence of humans in images showing their activities would be a strong cue of intangible dimension of heritage properties. Under such consideration, an Inception ResNet-V1 model (https://github.com/timesler/facenet-pytorch, accessed on 9 March 2022) pre-trained on the *VGGFace2* Dataset [73,74] has been used to generate features about depicted faces in the images. A three-dimensional vector $\mathfrak{f}_i$ was obtained for any image input $\mathfrak{I}_i$, where the non-negative first entry $\mathfrak{f}_{1,i} \in \mathbb{N}$ counts the number of faces detected in the image, the second entry $\mathfrak{f}_{2,i} \in [0, 1]$ records the confidence of the model for face detection, and the third entry $\mathfrak{f}_{3,i} \in [0, 1]$ calculates the proportion of area of all the bounding boxes of detected faces to the total area of the image. Similarly, the vectorized format could be written as $\boldsymbol{F} := [\mathfrak{f}_i]_{3 \times K}$ over the entire dataset.

Finally, all obtained visual features were concatenated vertically to generate the final visual feature $\boldsymbol{X}_{982 \times K}^{\mathrm{vis}}$:

$$\boldsymbol{X}_{982 \times K}^{\mathrm{vis}} = \left[ \boldsymbol{H}^{\mathrm{v}\mathsf{T}}, \boldsymbol{F}^{\mathsf{T}}, \sigma^{(5)}(\boldsymbol{L}^{\mathrm{s}})^{\mathsf{T}}, \sigma^{(10)}(\boldsymbol{L}^{\mathrm{a}})^{\mathsf{T}} \right]^{\mathsf{T}}, \tag{6}$$

where $[\cdot, \cdot]$ denotes the horizontal concatenation of matrices.

This final matrix is to be used in future MML tasks as the vectorized descriptor of the uni-modal visual contents of the posts, with both more abstract hidden features only to be understood by machines, and more specific information about predicted categories interpretable by humans, which is common practice in MML literature [13]. All models are tested on both 150 × 150 and 320 × 240 px images to compare the consistency of generated features. The workflow of generating visual features is illustrated in the top part of Figure 2.

### 2.4.2. Textual Features

In the last decade, attention- and Transformer-based pre-trained models have taken over the field of Natural Language Processing (NLP), increasing the performance of models in both general machine learning tasks, and domain-specific transfer-learning scenarios [75]. As an early version, the pre-trained Bidirectional Encoder Representations from Transformers (BERT) [76] is still regarded as a powerful base model to be fine-tuned on specific downstream datasets and for various NLP tasks. Specifically, the output on the [CLS] token of BERT models is regarded as an effective representation of the entire input sentence, being used extensively for classification tasks [77,78]. In the heritage studies domain, Bai et al. [79] fine-tuned BERT on the dataset *WHOSe Heritage* that they constructed from the UNESCO World Heritage inscription document, followed by a Multi-Layer Perceptron (MLP) classifier to predict the OUV selection criteria that a sentence is concerned with, showing top-1 accuracy of around 71% and top-3 accuracy of around 94%.

This study used the openly-released BERT model fine-tuned on *WHOSe Heritage* with PyTorch (https://github.com/zzbn12345/WHOSe_Heritage, accessed on 10 March 2022). The BERT model took both the entire sentence sets $\mathcal{S}_i$ and individual sentences of the sets $\{ \int_i^{(1)}, \int_i^{(2)}, \ldots, \int_i^{(|\mathcal{S}_i|)} \}$ as paragraph- and sentence-level inputs, respectively, for the comparison of consistency on predicted outputs of this new dataset. Furthermore, taking the entire sentence sets $\mathcal{S}_i$ as input, the 768-dimensional output vector $h_{768 \times 1}^{\text{BERT}}$ of the [CLS] token was retrieved on samples that have valid textual data:

$$h_i^{\text{BERT}} = f_{\text{BERT}}(\mathcal{S}_i | \Theta_{\text{BERT}}), \text{ where } f_{\text{BERT}}(\varnothing | \Theta_{\text{BERT}}) = \mathbf{0}_{768 \times 1}, \tag{7}$$

or preferably in a vectorized format:

$$H^{\text{B}} = f_{\text{BERT}}([\mathcal{S}_1, \mathcal{S}_2, \ldots, \mathcal{S}_K] | \Theta_{\text{BERT}}), \text{where } H^{\text{B}} := [h_i^{\text{BERT}}]_{768 \times K}. \tag{8}$$

Moreover, the original language of each sentence may provide additional information to the verbal context of posts, informative to effectively identify and compare locals and tourists. A three-dimensional vector $\mathbf{o}_i \in \{0, 1\}^3$ was obtained with Google Translator API. The three entries, respectively, marked whether there were sentences in English, local languages (Dutch, Chinese, or Italian, respectively), and other languages in the set $\mathcal{S}_i$. The elements of vector $\mathbf{o}_i$ or the matrix form $O := [\mathbf{o}_i]_{3 \times K}$ could be in a range from all zeros (when there were no textual data at all) to all ones (where the post comprised different languages in separate sentences).

Similar to visual features, final textual features $X_{771 \times K}^{\text{tex}}$ could be obtained as:

$$X_{771 \times K}^{\text{tex}} = \left[ H^{\text{B}^{\mathsf{T}}}, O^{\mathsf{T}} \right]^{\mathsf{T}}. \tag{9}$$

The workflow of generating textual features is illustrated in the bottom part of Figure 2.

### 2.4.3. Contextual Features

As mentioned in Section 2.3, the user ID $\mathfrak{u}_i$ and timestamp $\mathfrak{t}_i$ of a post are both an instance from their respective set $\mathcal{U}$ and $\mathcal{T}$, since multiple posts could be posted by the same user, and multiple images could be taken during the same week. To help formulate and generalize the problem under the practice of relational database [80], the information from both could be transformed as one-hot embeddings $U := [u_{j,i}]_{|\mathcal{U}| \times K} \in \{0, 1\}^{|\mathcal{U}| \times K}$ and $T := [t_{k,i}]_{|\mathcal{T}| \times K} \in \{0, 1\}^{|\mathcal{T}| \times K}$, such that:

$$u_{j,i} = \begin{cases} 1 & \text{if } \mathfrak{u}_i = \mu_j \in \mathcal{U} \\ 0 & \text{otherwise} \end{cases}, \tag{10}$$

$$\text{and } t_{k,i} = \begin{cases} 1 & \text{if } \mathfrak{t}_i = \tau_k \in \mathcal{T} \\ 0 & \text{otherwise} \end{cases}. \tag{11}$$

Furthermore, Section 2.3 and Appendix A also mentioned the collection of the public contacts and groups of all the users $\mu_j$ from the set $\mathcal{U}$. To keep the problem simple, only direct contact pairs were considered to model the back-end social structure of the users, effectively filtering out the other contacts a user $\mu_j$ had that were not in the set of interest $\mathcal{U}$, resulting in an adjacency matrix among the users $A^{\mathcal{U}} := [a^{\mathcal{U}}_{j,j'}]_{|\mathcal{U}| \times |\mathcal{U}|} \in \{0,1\}^{|\mathcal{U}| \times |\mathcal{U}|}, j, j' \in [1, |\mathcal{U}|]$ marking their direct friendship:

$$a^{\mathcal{U}}_{j,j'} = \begin{cases} 1 & \text{if } \mu_j \text{ and } \mu_{j'} \text{ are contacts or } j = j' \\ 0 & \text{otherwise} \end{cases}. \tag{12}$$

Let $\mathcal{I}(\mu_j)$ denote the set of public groups a user $\mu_j$ follows (can be an empty set if $\mu_j$ follows no group), and let $\text{IoU}(\mathcal{A}, \mathcal{B})$ denote the Jaccard Index (size of Intersection over size of Union) of two generic sets $\mathcal{A}, \mathcal{B}$:

$$\text{IoU}(\mathcal{A}, \mathcal{B}) = \frac{|\mathcal{A} \cap \mathcal{B}|}{|\mathcal{A} \cup \mathcal{B}| + \varepsilon}, \tag{13}$$

where $\varepsilon$ is a small number to avoid zero-division. Then another weighted adjacency matrix among the users could be constructed: $A^{\mathcal{U}'} := [a^{\mathcal{U}'}_{j,j'}]_{|\mathcal{U}| \times |\mathcal{U}|} \in [0,1]^{|\mathcal{U}| \times |\mathcal{U}|}, j, j' \in [1, |\mathcal{U}|]$, marking the mutual interests among the users as group subscription on Flickr:

$$a^{\mathcal{U}'}_{j,j'} = \text{IoU}(\mathcal{I}(\mu_j), \mathcal{I}(\mu_{j'})). \tag{14}$$

To further simplify the problem, although the geo-location $\mathfrak{l}_i = (\mathfrak{x}_i, \mathfrak{y}_i)$ of each post was typically distributed in a continuous 2D geographical space, it would be beneficial to further aggregate and discretize the distribution in a topological abstraction of spatial network [34,35,81], which has also been proven to be effective in urban spatial analysis, including but not limited to Space Syntax [82–85]. The OSMnx python library (https://osmnx.readthedocs.io/en/stable/, accessed on 28 February 2022) was used to inquire the simplified spatial network data on *OpenStreetMap* including all means of transportation [86] in each city with the same centroid location and radius described in Section 2.3. This operation effectively saved a spatial network as an undirected weighted graph $G_0 = (V_0, E_0, w_0)$, where $V_0 = \{v_1, v_2, \ldots, v_{|V_0|}\}$ is the set of all street intersection nodes, $E_0 \subseteq V_0 \times V_0$ is the set of all links possibly connecting two spatial nodes (by different sorts of transportation such as walking, biking, and driving), and $w_0 \in \mathbb{R}^{|E_0|}_+$ is a vector with the same dimension as the cardinality of the edge set, marking the average travel time needed between node pairs (dissimilarity weights). The `distance.nearest_nodes` method of OSMnx library was used to retrieve the nearest spatial nodes to any post location $\mathfrak{l}_i = (\mathfrak{x}_i, \mathfrak{y}_i)$. By only retaining the spatial nodes that bear at least one data sample posted nearby, and restricting the link weights between nodes so that the travel time on any link is no more than 20 min, which ensures a comfortable temporal distance forming neighbourhoods and communities [87], a subgraph $G = (V, E, w)$ of $G_0$ could be constructed, so that $V \subseteq V_0, E \subseteq E_0$, and $w \in [0, 20.0]^{|E|}$. As a result, another one-hot embedding matrix $S := [s_{l,i}]_{|V| \times K} \in \{0,1\}^{|V| \times K}$ could be obtained:

$$s_{l,i} = \begin{cases} 1 & \text{if the closest node to point } \mathfrak{l}_i \text{ is } v_l \in V \\ 0 & \text{otherwise} \end{cases}. \tag{15}$$

The contextual features constructed as matrices/graphs would be further used in Section 2.6 to link the posts together.

### 2.5. Pseudo-Label Generation

2.5.1. Heritage Values as OUV Selection Criteria

Various categories on heritage values (HV) have been provided by scholars [3,4,53,54]. To keep the initial step simple, this study arbitrarily applied the value definition in UNESCO WHL with regard to ten OUV selection criteria, as listed in Appendix Table A3 with an additional class Others representing scenarios where no OUV selection criteria suit the scope of a sentence (resulting in an 11-class category). It must be noted that the OUV selection criteria and the corresponding Statements of OUV include elements that could be identified and categorized as either heritage values or heritage attributes. Therefore, they are not necessarily heritage value per se, a detailed discussion on which falls out of the scope of this paper. However, for pragmatic purposes of demonstrating a framework, this study omits this distinction and considers the OUV selection criteria as a proxy of HV during label generation. A group of ML models were trained and fine-tuned to make such predictions by Bai et al. [79] as introduced in Section 2.4.2. Except for BERT already used to generate textual features as mentioned above, a Universal Language Model Fine-tuning (UMLFiT) [88] has also been trained and fine-tuned, reaching a similar performance in accuracy. Furthermore, it has been found that the average confidence by both BERT and ULMFiT models on the prediction task showed significant correlation with expert evaluation, even on social media data [79]. This suggests that it may be possible to use both trained models to generate labels about heritage values in a semi-supervised active learning setting [45,89], since this task is overly knowledge-demanding for crowd-workers, yet too time-consuming for experts [90].

The pseudo-label generation step could be formulated as:

$$y_i^{\text{BERT}} = \begin{cases} g_{\text{BERT}}(\mathcal{S}_i | \Theta_{\text{BERT}}) & \text{if } \mathcal{S}_i \neq \varnothing \\ \mathbf{0}_{11 \times 1} & \text{otherwise} \end{cases}, \tag{16}$$

$$y_i^{\text{ULMFiT}} = \begin{cases} g_{\text{ULMFiT}}(\mathcal{S}_i | \Theta_{\text{ULMFiT}}) & \text{if } \mathcal{S}_i \neq \varnothing \\ \mathbf{0}_{11 \times 1} & \text{otherwise} \end{cases}, \tag{17}$$

$$\mathbf{Y}^{\text{HV}} := [y_i^{\text{HV}}]_{11 \times K}, y_i^{\text{HV}} = \frac{y_i^{\text{BERT}} + y_i^{\text{ULMFiT}}}{2}. \tag{18}$$

where $g_{(*)}$ is an end-to-end function including both pre-trained models and MLP classifiers; and $y_i^{(*)}$ is an 11-dimensional logit vector as soft-label predictions. Let $\text{argmx}(l, n)$ denote the function returning the index set of the largest $n$ elements of a vector $l$, together with the previously defined $\max(l, n)$, the confidence and [dis-]agreement of models for top-$n$ predictions could be computed as:

$$\mathbf{K}^{\text{HV}} := [\boldsymbol{\kappa}_i^{\text{HV}}]_{2 \times K}, \boldsymbol{\kappa}_i^{\text{HV}} := [\kappa_i^{\text{HV}(0)}, \kappa_i^{\text{HV}(1)}]^{\mathsf{T}}, \tag{19}$$

$$\kappa_i^{\text{HV}(0)} = \sum_{n_0=1}^{n} \frac{\max(y_i^{\text{BERT}}, n_0) + \max(y_i^{\text{ULMFiT}}, n_0)}{2}, \tag{20}$$

$$\kappa_i^{\text{HV}(1)} = \text{IoU}(\text{argmx}(y_i^{\text{BERT}}, n), \text{argmx}(y_i^{\text{ULMFiT}}, n)). \tag{21}$$

This confidence indicator matrix $\mathbf{K}^{\text{HV}}$ could be presumably regarded as a filter for the labels on heritage values $\mathbf{Y}^{\text{HV}}$, to only keep the samples with high inter-annotator (model) agreement [46] as the "ground-truth" [pseudo-] labels, while treating the others as unlabeled [49,91].

2.5.2. Heritage Attributes as Depicted Scenery

Heritage attributes (HA) also have multiple categorization systems [5,26,92–94], and are arguably more vaguely defined than HV. For simplicity, this study arbitrarily combined the attribute definitions of Veldpaus [5] and Ginzarly et al. [26], and kept a 9-class category of tangible and/or intangible attributes visible from an image. More precisely, this category

should be framed as "depicted scenery" of an image [26] that heritage attributes could possibly be induced from. The depicted scenes themselves are not yet valid heritage attributes. This semantic/philosophical discussion, however, is out of the scope of this paper. The definitions of the nine categories are listed in Appendix Table A4.

An image dataset collected in Tripoli, Lebanon and classified with expert-based annotations presented by Ginzarly et al. [26] was used to train a few ML models to replicate the experts' behaviour on classifying depicted scenery with Scikit-learn python library [95]. For each image, a unique class label was provided, effectively forming a multi-class classification task. The same 512-dimensional visual representation $\boldsymbol{H}^V$ introduced in Section 2.4.1 was generated from the images as the inputs. Classifiers including Multi-layer Perceptron (MLP) (shallow neural network) [96], K-Nearest Neighbours (KNN) [97], Gaussian Naive Bayes (GNB) [98], Support Vector Machine (SVM) [99], Random Forest (RF) [100], and Bagging Classifier [101] with SVM core (BC-SVM) were first trained and tuned for optimal hyperparameters using 10-fold cross validation (CV) with grid search [102]. Then, the individually-trained models were put into ensemble-learning settings as both a voting [103] and a stacking classifier [104]. All trained models were tested on validation and test datasets to evaluate their performance. Details of the machine learning models are given in Appendix B. Both ensemble models were further applied in images collected in this study. Similar to the HV labels described in Section 2.5.1, the label generation step of HA could be formulated as:

$$y_i^{\text{VOTE}} = h_{\text{VOTE}}(h_i^V | \Theta_{\text{VOTE}}, \mathcal{M}, \boldsymbol{\Theta}_{\mathcal{M}}), \tag{22}$$

$$y_i^{\text{STACK}} = h_{\text{STACK}}(h_i^V | \Theta_{\text{STACK}}, \mathcal{M}, \boldsymbol{\Theta}_{\mathcal{M}}), \tag{23}$$

$$\boldsymbol{Y}^{\text{HA}} := [\boldsymbol{y}_i^{\text{HA}}]_{9 \times K}, \boldsymbol{y}_i^{\text{HA}} = \frac{y_i^{\text{VOTE}} + y_i^{\text{STACK}}}{2}. \tag{24}$$

where $\boldsymbol{h}_{(*)}$ is an ensemble model taking all parameters $\boldsymbol{\Theta}_{\mathcal{M}}$ from each ML model in set $\mathcal{M}$; and $\boldsymbol{y}_i^{(*)}$ is a 9-dimensional logit vector as soft-label predictions. Similarly, the confidence of models for top-$n$ prediction is:

$$\boldsymbol{K}^{\text{HA}} = [\boldsymbol{\kappa}_i^{\text{HA}}]_{2 \times K}, \boldsymbol{\kappa}_i^{\text{HA}} = [\kappa_i^{\text{HA}(0)}, \kappa_i^{\text{HA}(1)}]^T, \tag{25}$$

$$\kappa_i^{\text{HA}(0)} = \sum_{n_0=1}^{n} \frac{\max(\boldsymbol{y}_i^{\text{VOTE}}, n_0) + \max(\boldsymbol{y}_i^{\text{STACK}}, n_0)}{2}, \tag{26}$$

$$\kappa_i^{\text{HA}(1)} = \text{IoU}(\text{argmx}(\boldsymbol{y}_i^{\text{VOTE}}, n), \text{argmx}(\boldsymbol{y}_i^{\text{STACK}}, n)). \tag{27}$$

This matrix $\boldsymbol{K}^{\text{HA}}$ could also be regarded the filter for heritage attributes labels $\boldsymbol{Y}^{\text{HA}}$.

### 2.6. Multi-Graph Construction

Three types of similarities/ relations among posts were considered to compose the links connecting the post nodes: *temporal similarity* (posts with images taken during the same time period), *social similarity* (posts owned by the same people, by friends, and by people who share mutual interests), and *spatial similarity* (posts with images taken at the same or nearby locations). All three could be deduced from the contextual information in Section 2.4.3. As a result, an undirected weighted multi-graph (also known as Multi-dimensional Graph [44]) with the same node set and three different link sets could be constructed as $\mathcal{G} = (\mathcal{V}, \{\mathcal{E}^{\text{TEM}}, \mathcal{E}^{\text{SOC}}, \mathcal{E}^{\text{SPA}}\}, \{\boldsymbol{w}^{\text{TEM}}, \boldsymbol{w}^{\text{SOC}}, \boldsymbol{w}^{\text{SPA}}\})$, where $\mathcal{V} = \{v_1, v_2, \dots, v_K\}$ is the node set of all the posts, $\mathcal{E}^{(*)} \subseteq \mathcal{V} \times \mathcal{V}$ is the set of all links connecting two posts of one similarity type, and the weight vector $\boldsymbol{w}^{(*)} := [w_e^{(*)}]_{|\mathcal{E}^{(*)}| \times 1} \in \mathbb{R}_+^{|\mathcal{E}^{(*)}|}$ marks the strength of connections. The multi-graph $\mathcal{G}$ could also be easily split into three simple undirected weighted graphs $\mathcal{G}^{\text{TEM}} = (\mathcal{V}, \mathcal{E}^{\text{TEM}}, \boldsymbol{w}^{\text{TEM}})$, $\mathcal{G}^{\text{SOC}} = (\mathcal{V}, \mathcal{E}^{\text{SOC}}, \boldsymbol{w}^{\text{SOC}})$, and

$\mathcal{G}^{\text{SPA}} = (\mathcal{V}, \mathcal{E}^{\text{SPA}}, \boldsymbol{w}^{\text{SPA}})$ concerning each type of similarities. Each $\mathcal{G}^{(*)}$ corresponds to a weighted adjacency matrix $\boldsymbol{A}^{(*)} := [a_{i,i'}^{(*)}]_{K \times K} \in \mathbb{R}_+^{K \times K}, i, i' \in [1, K]$, such that:

$$
a_{i,i'}^{(*)} = \begin{cases} w_e^{(*)} & \text{if the } e_{\text{th}} \text{ element of } \mathcal{E} \text{ is } (v_i, v_{i'}), \\ 0 & \text{otherwise.} \end{cases} \tag{28}
$$

The three weighted adjacency matrices could be, respectively, obtained as follows:

### 2.6.1. Temporal Links

Let $\mathfrak{T}_{|\mathcal{T}| \times |\mathcal{T}|}$ denote a symmetric tridiagonal matrix where the diagonal entries are all 1 and off-diagonal non-zero entries are all $\alpha_{\mathcal{T}}$, where $\alpha_{\mathcal{T}} \in [0, 1)$ is a parametric scalar:

$$
\mathfrak{T}_{|\mathcal{T}| \times |\mathcal{T}|} := \begin{pmatrix} 1 & \alpha_{\mathcal{T}} & 0 & \cdots & 0 & 0 \\ \alpha_{\mathcal{T}} & 1 & \alpha_{\mathcal{T}} & \cdots & 0 & 0 \\ 0 & \alpha_{\mathcal{T}} & 1 & \cdots & 0 & 0 \\ \vdots & \vdots & \vdots & \ddots & \vdots & \vdots \\ 0 & 0 & 0 & \cdots & 1 & \alpha_{\mathcal{T}} \\ 0 & 0 & 0 & \cdots & \alpha_{\mathcal{T}} & 1 \end{pmatrix}, \tag{29}
$$

then the weighted adjacency matrix $\boldsymbol{A}_{K \times K}^{\text{TEM}}$ for temporal links could be formulated as:

$$
\boldsymbol{A}^{\text{TEM}} = \boldsymbol{T}^{\mathsf{T}} \mathfrak{T} \boldsymbol{T}, \boldsymbol{A}^{\text{TEM}} \in \{0, \alpha_{\mathcal{T}}, 1\}^{K \times K}, \tag{30}
$$

where $\boldsymbol{T}_{|\mathcal{T}| \times K}$ is the one-hot embedding of timestamp for posts mentioned in Equation (11). For simplicity, $\alpha_{\mathcal{T}}$ is set to 0.5. With such a construction, all posts from which the images were originally taken in the same week would have a weight of $w_e^{\text{TEM}} = 1$ connecting them in $\mathcal{G}^{\text{TEM}}$, and posts with images taken in nearby weeks in a chronological order would have a weight of $w_{e'}^{\text{TEM}} = 0.5$. Note, however, that the notion of "*nearby*" may not necessarily correspond to temporally adjacent weeks, as the interval of timestamps marking the date when a photo was taken could be months and even years in earlier time periods. In use cases sensitive to the time intervals, the value of $\alpha_{\mathcal{T}}$ could also be weighted: i.e., the longer the time interval actually is, the smaller $\alpha_{\mathcal{T}}$ becomes.

### 2.6.2. Social Links

Let $\mathfrak{U}_{|\mathcal{U}| \times |\mathcal{U}|}$ denote a symmetric matrix as a linear combination of three matrices marking the social relations among the users:

$$
\mathfrak{U}_{|\mathcal{U}| \times |\mathcal{U}|} = \frac{\alpha_{\mathcal{U}}^{(1)} \boldsymbol{I} + \alpha_{\mathcal{U}}^{(2)} \boldsymbol{A}^{\mathcal{U}} + \alpha_{\mathcal{U}}^{(3)} (\boldsymbol{A}^{\mathcal{U}'} > \beta_{\mathcal{U}})}{\alpha_{\mathcal{U}}^{(1)} + \alpha_{\mathcal{U}}^{(2)} + \alpha_{\mathcal{U}}^{(3)}}, \tag{31}
$$

where $\boldsymbol{I} \in \{0, 1\}^{|\mathcal{U}| \times |\mathcal{U}|}$ is a diagonal matrix of 1s for the *self relation*, $\boldsymbol{A}^{\mathcal{U}} \in \{0, 1\}^{|\mathcal{U}| \times |\mathcal{U}|}$ is the matrix mentioned in Equation (12) for the *friendship relation*, $(\boldsymbol{A}^{\mathcal{U}'} > \beta_{\mathcal{U}}) \in \{0, 1\}^{|\mathcal{U}| \times |\mathcal{U}|}$ is a mask on the matrix $\boldsymbol{A}^{\mathcal{U}'}$ introduced in Equation (14) for the *common-interest relation* above a certain threshold $\beta_{\mathcal{U}} \in (0, 1)$, and $\alpha_{\mathcal{U}}^{(1)}, \alpha_{\mathcal{U}}^{(2)}, \alpha_{\mathcal{U}}^{(3)} \in \mathbb{R}_+$ are parametric scalars to balance the weights of different social relations. The weighted adjacency matrix $\boldsymbol{A}_{K \times K}^{\text{SOC}}$ for social links could be formulated as:

$$
\boldsymbol{A}^{\text{SOC}} = \boldsymbol{U}^{\mathsf{T}} \mathfrak{U} \boldsymbol{U}, \boldsymbol{A}^{\text{SOC}} \in [0, 1]^{K \times K}, \tag{32}
$$

where $\boldsymbol{U}_{|\mathcal{U}| \times K}$ is the one-hot embedding of owner/user for posts mentioned in Equation (10). For simplicity, the threshold $\beta_{\mathcal{U}}$ is set to 0.05 and the scalars $\alpha_{\mathcal{U}}^{(1)}, \alpha_{\mathcal{U}}^{(2)}, \alpha_{\mathcal{U}}^{(3)}$ are all set to 1. With such a construction, all posts uploaded by the same user would have a weight of $w_e^{\text{SOC}} = 1$ connecting them in $\mathcal{G}^{\text{SOC}}$, posts by friends with common interests (of more



than 5% common groups subscriptions) would have a weight of $w_{e'}^{\text{SOC}} = \frac{2}{3}$, and posts by either friends with little common interests or strangers with common interests would have a weight of $w_{e''}^{\text{SOC}} = \frac{1}{3}$.

### 2.6.3. Spatial Links

Let $\mathfrak{S} := [\mathfrak{s}_{l,l'}] \in [0,1]^{|V| \times |V|}, l, l' \in [1, |V|]$ denote a symmetric matrix computed with simple rules showing the spatial closeness (conductance) of nodes from the spatial graph $G = (V, E, \boldsymbol{w})$ mentioned in Section 2.4.3, whose weights $\boldsymbol{w} := [w_e]_{|E| \times 1} \in [0, 20.0]^{|E|}$ originally showed the distance of nodes (resistance):

$$\mathfrak{s}_{l,l'} = \begin{cases} \frac{20 - w_e}{20} & \text{if the } e_{\text{th}} \text{ element of } E \text{ is } (v_l, v_{l'}), \\ 0 & \text{otherwise.} \end{cases} \tag{33}$$

The weighted adjacency matrix $A_{K \times K}^{\text{SPA}}$ for spatial links could be formulated as:

$$A^{\text{SPA}} = S^{\mathsf{T}} \mathfrak{S} S, A^{\text{SPA}} \in [0,1]^{K \times K}, \tag{34}$$

where $S_{|V| \times K}$ is the one-hot embedding of spatial location for posts mentioned in Equation (15). With such a construction, posts located at the same spatial node would have a weight of $w_e^{\text{SPA}} = 1$ in $\mathcal{G}^{\text{SPA}}$, and posts from nearby spatial nodes would have a weight linearly decayed based on distance within a maximum transport time of 20 min.

Additionally, the multi-graph $\mathcal{G}$ could be simplified as a simple composed graph $\mathcal{G}' = (\mathcal{V}, \mathcal{E}')$ with a binary adjacency matrix $A \in \{0,1\}^{K \times K}$, such that:

$$A := (A^{\text{TEM}} > 0) \vee (A^{\text{SOC}} > 0) \vee (A^{\text{SPA}} > 0), \tag{35}$$

which connects two nodes of posts if they are connected and similar in at least one contextual relationship.

All graphs were constructed with NetworkX python library [105]. The rationale under constructing various graphs was briefly described in Section 1: the posts close to each other (in temporal, social, or spatial contexts) could be arguably similar in their contents, and therefore, also similar in the heritage values and attributes they might convey. Instead of regarding these similarities as redundant and, e.g., removing duplicated posts by the same user to avoid biasing the analysis, such as in Ginzarly et al. [26], this study intends to take advantage of as much available data as possible, since similar posts may enhance and strengthen the information, compensating the redundancies and/or nuances using back-end graph structures. At later stages of the analysis, the graph of posts could be even coarsened with clustering and graph partitioning methods [44,106–108], to give an effective summary of possibly similar posts.

## 3. Analyses as Qualitative Inspection

### 3.1. Sample-Level Analyses of Datasets

#### 3.1.1. Generated Visual and Textual Features

Table 3 shows the consistency of generated visual and textual features. The visual features compared the scene and attribute predictions on images of difference sizes ($150 \times 150$ and $320 \times 240$ px); and the textual features compared the OUV selection criteria with aggregated (averaged) sentence-level predictions on each sentence from set $\{ \mathfrak{f}_i^{(1)}, \mathfrak{f}_i^{(2)}, \ldots, \mathfrak{f}_i^{(|\mathcal{S}_i|)} \}$ and paragraph-/post-level predictions on set $\mathcal{S}_i$.

For both scene and attribute predictions, the means of top-1 Jaccard Index were always higher than that of top-$n$, however, the smaller variance proved the necessity of using top-$n$ prediction as features. Note the attribute prediction was more stable than the scene prediction when the image shape changed, this is probably because the attributes usually describe low-level features which could appear in multiple parts in the image, while some critical information to judge the image scene may be lost during cropping and resizing

in the original ResNet-18 model. Considering the relatively high consistency of model performance and the storage cost of images when the dataset would ultimately scale up (e.g., VEN-XL), the following analyses would only be performed on smaller square images of 150 × 150 px.

The high Jaccard Index of OUV predictions showed that averaging the textual features derived from sub-sentences of a paragraph would yield a similar performance of directly feeding the whole paragraph into models, especially when the top-3 predictions are of main interest. Note that the higher consistency in Suzhou was mainly a result of the higher proportion of posts only consisting of one sentence.

**Table 3.** The consistency (the mean and standard deviation of top-*n* IoU Jaccard Index on predicted sets) of generated features. For visual features, predictions with different input image sizes (150 × 150 px and 320 × 240 px) are compared; for textual features, average sentence-level predictions and paragraph-/post-level predictions are compared. The best scores for each feature are in bold, and the selected ones for future tasks are underlined. "#" means "the number of" in the table.

| Sets to Calculate IoU Jaccard Index | AMS | SUZ | VEN |
|---|---|---|---|
| # Compared Posts w. Visual Features | 3727 | 3137 | 2951 |
| Top-1 scene predictions | **0.656** | **0.676** | **0.704** |
| —argmx $(l^{s}, 1)$ | (0.475) | (0.468) | (0.456) |
| **Top-5 scene predictions** | 0.615 | 0.636 | 0.635 |
| —argmx $(l^{s}, 5)$ | **(0.179)** | **(0.238)** | **(0.229)** |
| Top-1 attribute predictions | **0.867** | **0.853** | **0.838** |
| —argmx $(l^{a}, 1)$ | (0.339) | (0.354) | (0.368) |
| **Top-10 attribute predictions** | 0.820 | 0.802 | 0.819 |
| —argmx $(l^{a}, 10)$ | **(0.140)** | **(0.144)** | **(0.139)** |
| # Compared Posts w. Textual Features | 2904 | 754 | 1761 |
| Top-1 OUV predictions | 0.775 | 0.923 | 0.714 |
| — argmx $(y^{BERT}, 1)$ | (0.418) | (0.267) | (0.452) |
| **Top-3 OUV predictions** | **0.840** | **0.938** | **0.791** |
| —argmx $(y^{BERT}, 3)$ | **(0.246)** | **(0.182)** | **(0.266)** |

Table 4 gives descriptive statistics of results that were not compared against different scenarios as in Table 3. Only a small portion of posts had detected faces in them. While Amsterdam has the highest proportion of face pictures (17.9%), Venice has larger average area of faces on the picture (i.e., more self-taken photos and tourist pictures). These numbers are also assumed to help associate a post to human-activity-related heritage values and attributes. Considering the languages of the posts, Amsterdam showed a balance between Dutch-speaking locals and English-speaking tourists, Venice showed a balance between Italian-speaking people and non-Italian-speaking tourists, while Suzhou showed a lack of Chinese posts. This is consistent with the popularity of Flickr as social media in different countries, which also implies that data from other social media could compensate this unbalance if the provisional research questions would be sensitive to the nuance between local and tourist narratives.

### 3.1.2. Pseudo-Labels for Heritage Values and Attributes

As argued in Section 2.5.1, the label generation process of this paper did not involve human annotators. Instead, it used thoroughly trained ML models as machine replica of annotators and considered their confidences and agreements as a filter to maintain the "high-quality" labels as pseudo-labels. Similar operations can be found in semi-supervised learning [48,49,91].

**Table 4.** Descriptive statistics (mean and standard deviation or counts, respectively) of the facial recognition results ***F*** as visual features and original language ***O*** as textual features. "#" means "the number of" in the table.

| Features | AMS | SUZ | VEN | VEN-XL |
|---|---|---|---|---|
| # Posts w. Faces | 667 | 303 | 166 | 9287 |
| # Faces detected | 1.547 | 1.403 | 1.349 | 1.298 |
| —$\mathfrak{f}_1$ | (0.830) | (0.707) | (0.785) | (0.651) |
| Model Confidence | 0.955 | 0.956 | 0.930 | 0.948 |
| —$\mathfrak{f}_2$ | (0.079) | (0.081) | (0.099) | (0.081) |
| Area proportion of faces | 0.049 | 0.057 | 0.077 | 0.076 |
| —$\mathfrak{f}_3$ | (0.112) | (0.073) | (0.185) | (0.112) |
| # Posts w. Texts * | 2904 | 754 | 1761 | 49,823 |
| # Posts in English $\mathfrak{o}_1$ | 1488 | 368 | 640 | 20,271 |
| # Posts in Native Lang $\mathfrak{o}_2$ | 1773 | 27 | 1215 | 28,633 |
| # Posts in Other Lang $\mathfrak{o}_3$ | 536 | 413 | 657 | 21,916 |

\* Note this is smaller than the sum of the three below, since each post can be written in multiple languages.

For heritage values, an average top-3 confidence of $\kappa^{\mathrm{HV}(0)} > 0.75$ and top-3 agreement (Jaccard Index) of $\kappa^{\mathrm{HV}(1)} > 0.5$ was used as the filter for $\mathbf{Y}^{\mathrm{HV}}$. This resulted in around 40–50% of the samples with textual data in each city as "labelled", and the rest as "unlabelled". Figure 3 demonstrates the distribution of "labelled" data about heritage values in each city. For all cities, cultural values are far more frequent than natural values, consistent with their status of cultural WH. However, elements related to natural values could still be found and were mostly relevant. The actual OUV inscribed in WHL mentioned in Table 1 could all be observed as significantly present (e.g., criteria (i),(ii),(iv) for Amsterdam) except for criterion (v) in Venice and Suzhou, which might be caused by the relatively fewer examples and poorer class-level performance of criterion (v) in the original paper [79]. Remarkably, criterion (iii) in Amsterdam and criterion (vi) in Amsterdam and Suzhou were not officially inscribed, but appeared to be relevant inducing from social media, inviting further heritage-specific investigations. The distributions of Venice and Venice-large were more similar in sentence-level predictions (Kullback–Leibler Divergence $D_{\mathrm{KL}} = 0.002$, Chi-square $\chi^2 = 39.515$) than post-level ($D_{\mathrm{KL}} = 0.051, \chi^2 = 518.895$), which might be caused by the specific set of posts sub-sampled in the smaller dataset.

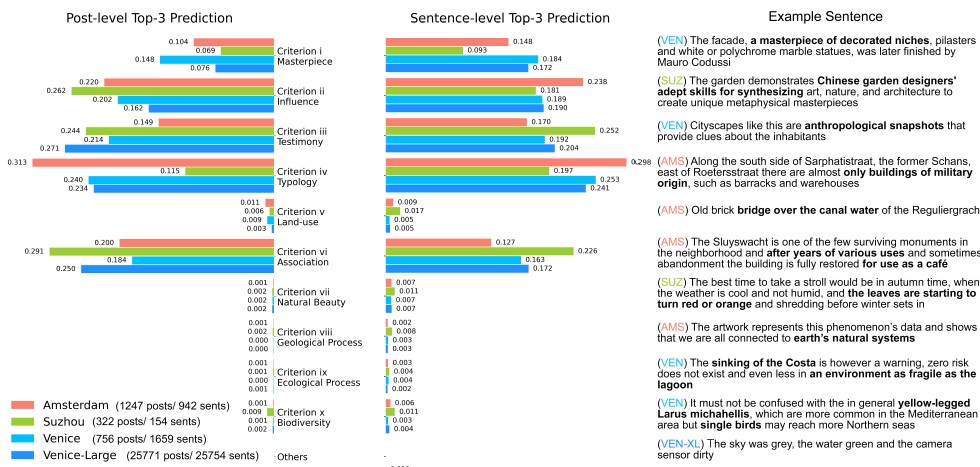

**Figure 3.** The proportion of posts and sentences that are predicted and labeled as each heritage value (OUV selection criterion) as top-3 predictions by both BERT and ULMFiT. One typical sentence from each category is also given in the right part of the figure.

For heritage attributes, Table 5 shows the performance of ML models mentioned in Section 2.5.2 and Appendix B. The two ensemble models with voting and stacking settings

performed equally well and significantly better than other models (except for CV accuracy of SVM), proving the rationale of using both classifiers for heritage attribute label prediction. An average top-1 confidence of $\kappa^{HA(0)} > 0.7$ and top-1 agreement of $\kappa^{HA(1)} = 1$ was used as the filter for $Y^{HA}$. This filter resulted in around 35–50% of the images in each city as "labelled", and the rest as "unlabelled". Figure 4 demonstrates the distribution of "labelled" data about heritage attributes in each city. It is remarkable that although the models were only trained on data from Tripoli, they performed reasonably well in unseen cases of Amsterdam, Suzhou, and Venice, capturing typical scenes of monumental buildings, architectural elements, and gastronomy, etc., respectively. Although half of the collected images were treated as "unlabelled" due to low confidence, the negative examples are not necessarily incorrect (e.g., with *Monuments and Buildings*). For all cities, *Urban Form Elements* and *People's Activity and Association* are the most dominant classes, consistent with the fact that most Flickr images are taken on the streets. Seen from the bar plots in Figure 4, the classes were relatively unbalanced, suggesting that more images from small classes might be needed or at least augmented in future applications. Furthermore, the distributions of Venice and Venice-large are similar to each other ($D_{KL} = 0.076$, $\chi^2 = 188.241$), suggesting a good representativeness of the sampled small dataset.

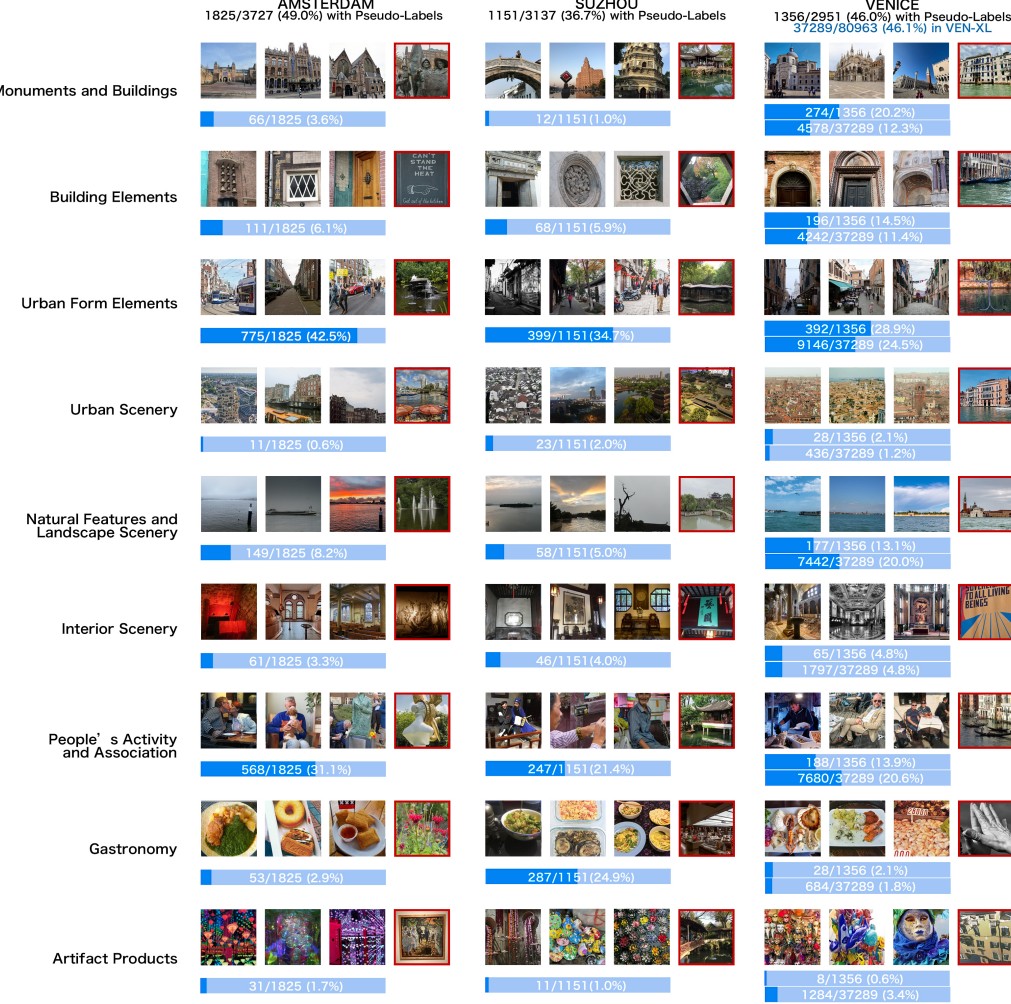

**Figure 4.** Typical image examples in each city labelled as each heritage attribute category (depicted scene) and bar plots of their proportions in the datasets (length of bright blue background bars represent 50%). Three examples with high confidence and one negative example with low confidence (in red frame) are given. All images are 150 × 150 px "thumbnails" flagged as "downloadable".

**Table 5.** The performance of models during the cross validation (CV) parameter selection, on the validation set, and on the test set of data from Tripoli. The best two models for each performance are in bold typeface, and the best underlined.

| ML Model | CV Acc | Val Acc | Val F1 | Test Acc | Test F1 |
|----------|--------|---------|--------|----------|---------|
| MLP | 0.767 | 0.749 | 0.70 | 0.789 | 0.72 |
| KNN | 0.756 | 0.724 | 0.67 | 0.767 | 0.71 |
| GNB | 0.738 | 0.749 | 0.71 | 0.800 | 0.77 |
| SVM | **0.797** | 0.754 | 0.71 | 0.822 | 0.78 |
| RF | 0.766 | 0.734 | 0.68 | 0.789 | 0.72 |
| BC-SVM | 0.780 | 0.759 | 0.71 | 0.811 | 0.74 |
| **VOTE** | 0.788 | **0.764** | **0.72** | **0.855** | **0.82** |
| **STACK** | 0.794 | **0.768** | **0.72** | 0.844 | 0.81 |

### 3.2. Graph-Level Analyses of Datasets

#### 3.2.1. Back-End Geographical Network

The back-end spatial structures of post locations as graphs $G = (V, E, \boldsymbol{w})$ were visualized in Figure 5. Further graph statistics in all cities were given in Table 6. The urban fabric is more visible in Venice than the other two cities, as there is always a dominant large component connecting most nodes in the graph, leaving fewer unconnected isolated nodes alone. While in Amsterdam, more smaller connected components exist together with a large one; and in Suzhou, the graph is even more fragmented with smaller components. This is possibly related to the distribution of tourism destinations, collectively forming bottom-up tourism districts (or "*tourist city*" as proposed in Encalada-Abarca et al. [109]), which is also consistent with the zoning typology of WH property concerning urban morphology [51,52]: for Venice, the Venetian islands are included together with a larger surrounding lagoon in the WH property (formerly referred to as core zone), and are generally regarded as a tourism destination as a whole; for Amsterdam, the WH property is only a part of the old city being mapped where tourists can freely wander and take photos in areas not listed yet as interesting tourism destinations; while for Suzhou, the WH properties are themselves fragmented gardens distributed in the old city, also representing the main destinations visited by (foreign) tourists.

**Table 6.** The statistics for the back-end Geographical Network $G = (V, E, \boldsymbol{w})$. "#" means "the number of" in the table.

| Graph Features | AMS | SUZ | VEN | VEN-XL |
|----------------|-----|-----|-----|--------|
| # Nodes in $V$ | 788 | 230 | 915 | 3549 |
| # Edges in $E$ | 3331 | 680 | 10,385 | 120,033 |
| # Connected Components | 72 | 38 | 6 | 13 |
| # Nodes Largest CC * | 355 | 50 | 897 | 3498 |
| Graph Density | 0.011 | 0.026 | 0.025 | 0.019 |
| # Isolated Nodes in $V_0 \backslash V$ | 157 | 88 | 20 | 22 |

* Connected Components.

Furthermore, the two types of rank-size plots showing, respectively, the degree distribution and the posts-per-node distribution revealed similar patterns, the latter being more heavy-tailed, a typical characteristic of large-scale complex networks [40,110], while the back-end spatial networks are relatively more regular.

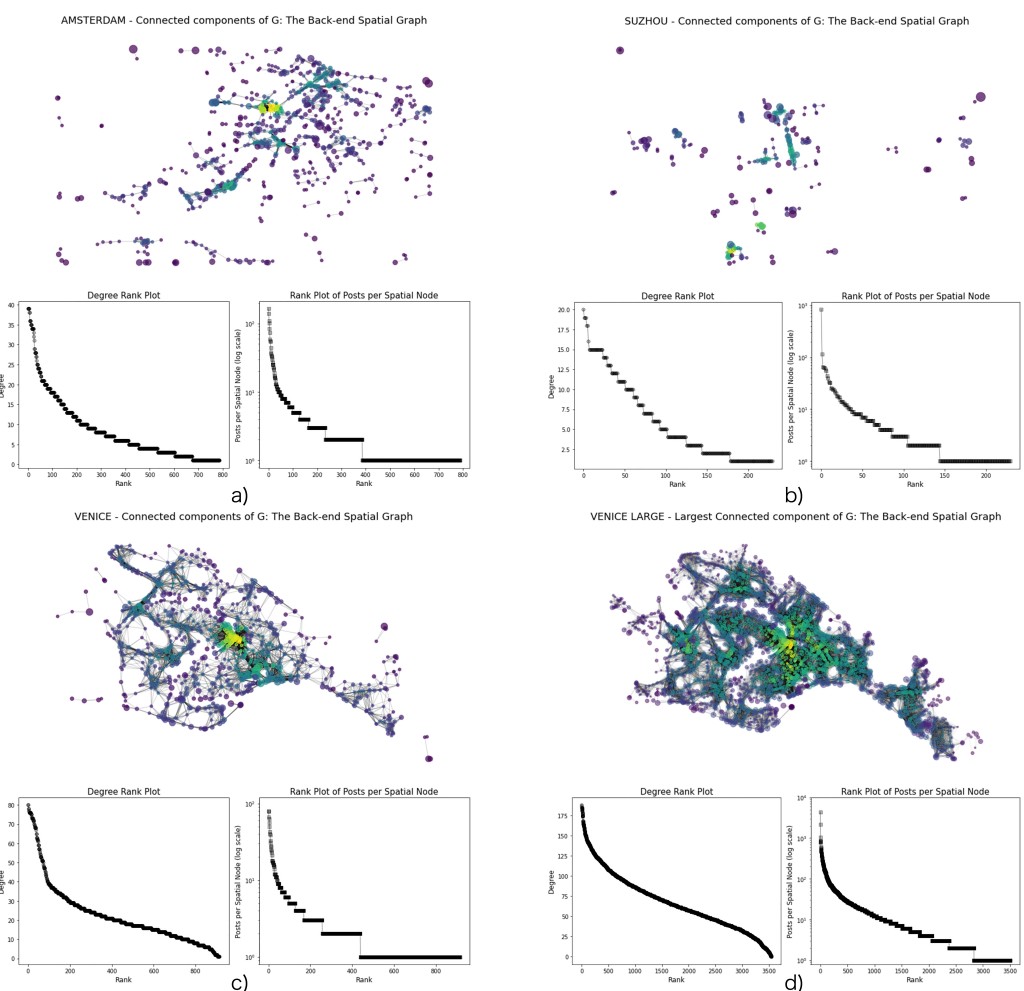

**Figure 5.** The back-end geographical networks for three case studies, respectively, showing the graph structure, degree ranking distribution, and the ranking distribution of posts per geo-spatial node (on a logarithm scale) in Amsterdam, Suzhou, Venice, and Venice-XL. The sizes of nodes denote the number of nearby posts allocated to the nodes, and the colors of nodes illustrate the degree of the node on the graph. Each link connects two nodes reachable to each other within 20 min.

### 3.2.2. Multi-Graphs and Sub-Graphs of Contextual Information

Table 7 shows graph statistics of three constructed sub-graphs $\mathcal{G}^{\text{TEM}}, \mathcal{G}^{\text{SOC}}, \mathcal{G}^{\text{SPA}}$ with different link types within the multi-graph $\mathcal{G}$, and the simple composed graph $\mathcal{G}'$ for each city, while Figure 6 plots their [weighted] degree distributions, respectively. The multi-graphs are further visualized in Appendix Figure A1.

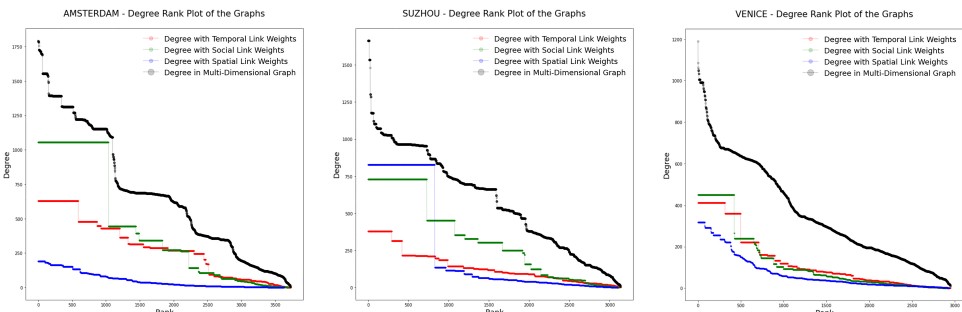

**Figure 6.** The rank-size plots of the degree distributions in the three cases of Amsterdam, Suzhou, and Venice, with regard to the temporal links, social links, spatial links, as well as the entire multi-graph.

**Table 7.** The statistics for the multi-graphs. "#" means "the number of" in the table.

| Graph Features | AMS | SUZ | VEN |
|---|---|---|---|
| Temporal Graph $\mathcal{G}^{\text{TEM}} = (\mathcal{V}, \mathcal{E}^{\text{TEM}}, w^{\text{TEM}})$ | | | |
| # Nodes * | 3727 | 3137 | 2951 |
| # Edges | 692,839 | 293,328 | 249,120 |
| Diameter | 145 | 116 | 270 |
| Graph Density | 0.100 | 0.060 | 0.057 |
| Social Graph $\mathcal{G}^{\text{SOC}} = (\mathcal{V}, \mathcal{E}^{\text{SOC}}, w^{\text{SOC}})$ | | | |
| # Nodes ** | 3696 | 3120 | 2916 |
| # Edges | 877,584 | 602,821 | 242,576 |
| # Connected Components | 47 | 56 | 60 |
| # Nodes Largest CC | 2694 | 942 | 2309 |
| Diameter Largest CC | 7 | 6 | 10 |
| Graph Density | 0.129 | 0.124 | 0.057 |
| Spatial Graph $\mathcal{G}^{\text{SPA}} = (\mathcal{V}, \mathcal{E}^{\text{SPA}}, w^{\text{SPA}})$ | | | |
| # Nodes ** | 3632 | 3102 | 2938 |
| # Edges | 135,079 | 415,049 | 221,414 |
| # Connected Components | 134 | 91 | 13 |
| # Nodes Largest CC | 1485 | 829 | 2309 |
| Diameter Largest CC | 22 | 1 | 22 |
| Graph Density | 0.020 | 0.086 | 0.051 |
| Simple Composed Graph $\mathcal{G}' = (\mathcal{V}, \mathcal{E}')$ | | | |
| # Nodes * | 3727 | 3137 | 2951 |
| # Edges | 1,271,171 | 916,496 | 534,513 |
| Diameter | 4 | 5 | 4 |
| Graph Density | 0.183 | 0.186 | 0.123 |

* By definition a connected graph (only one connected component). ** The isolated nodes with no links are not counted here, therefore the numbers of nodes are smaller than the actual size of the node set $\mathcal{V}$.

The three link types provided heterogeneous characteristics: (1) the temporal graph is by definition connected, where the highest density in Amsterdam suggested the largest number of photos taken in consecutive time periods, while the largest diameter in Venice suggested the broadest span of time; (2) the social graph is structured by the relationship of users, where the largest connected components showed clusters of posts shared either by the same user, or by users who are friends or with mutual interests, the size of which in Suzhou is small because of the fewest users shown in Table 1; (3) the spatial graph shows a similar connectivity pattern with the back-end spatial networks/graphs, where the extremely small diameter and the largest density in Suzhou reassured the fragmented positions of posts; (4) although the degree distribution of three sub-graphs fluctuated due to the different socio-economic and spatio-temporal characteristics of different cities, that of the simple composed graph showed similar elbow-shaped patterns, with similar density and diameter. Moreover, the heterogeneous graph structures suggest that different parameters and/or backbone models need to be fit and fine-tuned with each link type, a common practice for deep learning on multi-graphs.

As the heterogeneous characteristics of constructed multi-graphs in the three cities are shown to be logically correspondent to reality, substantiating the generality of the methodological framework, they could be used as contextual information to aid future semi-supervised classification tasks concerning heritage values and attributes.

## 4. Discussion

### 4.1. Provisional Tasks for Urban Data Science

The datasets introduced could be used to answer questions from the perspectives of machine learning and social network analysis as well as heritage and urban studies. Table 8 gives a few provisional tasks that could be realised using the collected datasets of this paper, and further datasets to be collected using the same introduced framework.

These problems would use some or all of extracted features (visual, textual, contextual), generated labels (heritage values and attributes), constructed graph structures, and even raw data as input and output components to find the relationship function among them.

Some problems are more interesting as ML/SNA problems (such as 4, 7 and 8), some are more fundamental for heritage studies and urban data science (such as 0, 1 and 6). While the former tends towards the technical and theoretical end of the whole potential range of the datasets, the latter tends towards the application end. However, to reach a reasonable performance during applications and discoveries, as is the main concern and interest for urban data science, further technical investigations and validations would be indispensable.

Even before performing such provisional tasks with the datasets created using the proposed framework in this study, the dataset creation and qualitative inspection process can already reveal interesting facts related to heritage studies, though they are performed primarily to check the quality of the created datasets in terms of their coherence and consistency. The analyses shown in Section 3.1.2 about the pseudo-labels generated for the topics of heritage values and attributes provide the most trivial contribution to cultural heritage studies. On the one hand, it demonstrates that the proposed framework could transfer knowledge from pre-trained models and provide meaningful predictions as replica of authoritarian views to justify heritage values and attributes. On the other hand, the distribution of generated labels both give an expected (as examples visualized in Figures 3 and 4) and unexpected (for example the significant appearance of OUV selection criteria originally not inscribed in WHL) outcomes that could invite further heritage investigations. The analyses of generated features shown in Section 3.1.1, however, could also provide strong clues informative to heritage studies. As argued in Section 2.4, both machine-readable abstract features such as hidden-layer vectors and human-interpretable prediction categories are stored as multi-modal features. While conducting future machine learning training, sensitivity checks on such human-interpretable features could give insights on how and what the models learn. For example, one would expect a model predicting the heritage value of "*criterion (vi)—association*" and heritage attributes of "*People's Activity and Association*" to pay much attention to the number and proportion of human faces in the image, *and vice versa*, hence the extraordinary appearance of both categories in the city of Amsterdam. As for the graph analyses in Section 3.2, while providing a basis for further graph-based semi-supervised learning of similar posts in nearby places, from the same time period, and by alike social groups, the spatio-temporal and socio-economic distribution of posts (as a proxy to social behaviour) already tells a story. For instance, as has been extensively argued by researchers such as Bill Hillier et al., one can often find a clear correspondence between the "buzz" or vitality of human activities in cities with the inherent centrality distributions on the network representation of the underlying space [82–84]. The co-appearance of large circles (large number of posts, thus high vitality) and warm colours (high centrality), and the visible clustering of warms colours (around places with good connectivity, such as the *Rialto* bridge and *San Marco* in Venice, confirming the conclusions drawn by Psarra [111]) shown in Figure 5 could further demonstrate such findings. Culturally significant locations are often important not only due to their individual attributes but also due to the embedding in their contexts, which inevitably renders cultural heritage studies contextual.

Further advanced analyses for directly answering domain-specific questions in cultural heritage studies (such as Questions 3 and 8 about the mechanism of contextual influence of posts to the mapping, extraction, and inference of heritage values and attributes) have been categorized in Table 8. Note that a further distinction needs to be made within the extracted heritage values and attributes, as they may essentially be clustered into three categories: (1) core heritage values and attributes officially listed and recognized that thoroughly define the heritage status; (2) values and attributes relevant to conservation and preservation practice; (3) other values and attributes not specifically heritage-related yet are conveyed to the same heritage property by ordinary people. This distinction should be made clear for practitioners intending to make planning decisions based on the conclusions drawn from studying such datasets.

**Table 8.** A few provisional tasks with formal problem definitions that could be performed. Potential scientific and social relevance for the Machine Learning community, and urban and/or heritage researchers, respectively, are given. The gray texts in the third column give a high-level categorization for each specific type of task in the context of machine learning.

| ID | Problem Definition | Type of Task | As a *Machine Learning /Social Network Analysis* Problem | As an *Urban/Heritage Study* Question |
|---|---|---|---|---|
| 0 | $X^{\mathrm{vis}} \mapsto Y^{\mathrm{HV}}|K^{\mathrm{HV}}$ | Image Classification *(semi-supervised)* | Using visual features to infer categories originally induced from (possibly missing) texts with co-training [112] in few-shot learning settings [113]. | As the latest advances in heritage value assessment have been discovering the added value of inspecting texts [4], can values also be seen and retrieved from scenes of images? |
| 1 | $X^{\mathrm{tex}} \mapsto Y^{\mathrm{HA}}|K^{\mathrm{HA}}$ | Text Classification *(semi-supervised)* | Using textual features to infer categories originally induced from images possibly with attention mechanisms [75]. | How to relate the textual descriptions to certain heritage attributes [28]? Are there crucial hints other than appeared nouns? |
| 2 | $X := \left\{X^{\mathrm{vis}}, X^{\mathrm{tex}}\right\} \mapsto Y := \left\{Y^{\mathrm{HV}}|K^{\mathrm{HV}}, Y^{\mathrm{HA}}|K^{\mathrm{HA}}\right\}$ | Multi-modal Classification *(semi-supervised)* | Using multi-modal (multi-view) features to make inference, either with training joint representations or by making early and/or late fusions [13,112]. | How can heritage values and attributes be jointly inferred from the combined information of both visual scenes and textual expressions [26]? How can they complement each other? |
| 3 | $X, A \mapsto Y$ | Node Classification *(semi-supervised)* | Test-beds for different graph filters such as Graph Convolution Networks [50] and Graph Attention Networks [114]. | How can the contextual information of a post contribute to the inference of its heritage values and attributes? What is the contribution of time, space, and social relations [115]? |
| 4 | $X, Y, A \mapsto A + A_{\mathrm{new}}$ | Link Prediction and Recommendation System *(semi-supervised)* | Test-beds for link prediction algorithms [116] considering current graph structure and node features. What is the probability that other links also should exist? | Considering the similarity of posts, would there be heritage values and attributes that also suit the interest of another user, fit another location, and/or reflect another period of time [117]? |
| 5 | $X, Y, A \mapsto \hat{X}, \hat{Y}, \hat{A}$ | Graph Coarsening *(unsupervised)* | Test-beds for graph pooling [44] and graph partitioning [106] algorithms to generate coarsened graphs [118] in different resolutions. | How can we summarize, aggregate, and eventually visualize the large-scale information from the social media platforms based on their contents and contextual similarities [30]? |
| 6 | $X, Y, A \mapsto y_{\mathcal{G}}^{\mathrm{HV}}|Y^{\mathrm{HV}}, y_{\mathcal{G}}^{\mathrm{HA}}|Y^{\mathrm{HA}}$ | Graph Classification *(supervised)* | Test-beds for graph classification algorithms [119] when more similar datasets have been collected and constructed in more case study cities. | Can we summarize the social media information of any city with World Heritage property so that the critical heritage values and attributes could be directly inferred [25]? |
| 7 | $X, Y, A \mapsto \mathfrak{I}, \mathcal{S}$ | Image/Text Generation *(supervised)* | Using multi-modal features to generate the missing and/or unfit images and/or textual descriptions, probably with Generative Adversarial Network [120]. | How can a typical image and/or textual description of certain heritage values and attributes at a certain location in a certain time by a certain type of user in a specific case study city be queried or even generated [28]? |

**Table 8.** *Cont.*

| ID | Problem Definition | Type of Task | As an *ML/SNA* Problem | As an *Urban/Heritage Study* Question |
|---|---|---|---|---|
| 8 | $X, Y, A^{\text{TEM}}, A^{\text{SOC}}, A^{\text{SPA}} \mapsto R + R^{\text{TEM}} + R^{\text{SOC}} + R^{\text{SPA}}$ | Attributed Multi-Graph Embedding *(self-supervised)* | Respectively generating a universal embedding and a context-specific embedding for each type of links in the multi-dimensional network [121], probably with random walks on graphs. | How are heritage values and attributes distributed and diffused in different contexts? Is the First Law of Geography [32] still valid in the specific social, temporal and spatial graphs? |
| 9 | $X^{(k)}, Y^{(k)}, A^{(k)}, T \mapsto X^{(k+1)}, Y^{(k+1)}, A^{(k+1)}$ | Dynamic Prediction *(self-supervised)* | Given the current graph structure and its features stamped with time steps, how shall it further evolve in the next time steps [36,122]? | How are the current expressions of heritage values and attributes in a city influencing the emerging post contents, the tourist behaviours, and the planning decision making [9,37]? |

One advantage of the proposed framework is that it allows for the creation of multigraphs from multiple senses of proximity or similarity in geographical, temporal, and/or the social space. In cases where one cannot easily find a ground truth, i.e., in exploratory analyses, having the possibility to treat the dataset as a set of connected data points instead of a powder-like set will be advantageous. The sense of similarity between data points by virtue of geographical/spatial proximity is arguably the oldest type of connection between them. However, when there is no exact physical sense of proximity in a geographical space, or when other forms of connection, e.g., through social media, are of influence, data scientists can benefit from other clues such as temporal connections related to the events or the social connections related to community structures. These can all inform potential questions to be answered in future studies.

Moreover, after retrieving knowledge of heritage values and attributes in case study cities from multi-modal UGC, for the sake of visualization, assessment and comparison during decision-making processes, further bundling and aggregation of individual data points would be desirable, as was briefly mentioned in Section 1 and also formulated in Table 8 as Question 5. This could be performed with all three proposed contextual information types denoting the proximity of data points. Data bundling and aggregation in the spatial domain would be the first action for creating a map. Depending on different use cases, this could be performed either on scale-dependent representations of geographical/administrative units (such as the natural islands divided by canals, or the so-called *parish islands/communities* in Venice [111]), or on identified clusters based on regular grids at different scales (such as the "*tourism districts*" [109]). While the use of the former (i.e., top-down boundaries) is trivial for administrative purposes, the latter (i.e., bottom-up clusters) could be arguably more generalizable in other cases, reflecting a universal collective sense of place [109]. Data bundling and aggregation in the temporal domain would map the generated features and labels on a discrete timeline at different scales (e.g., months, years, decades, etc.), presumably of sufficiently high resolution to capture the temporal dynamics and variations of data. For example, one may find that some topics are extensively mentioned in only a short period of time, while others pertain for longer spans, suggesting different patterns of public perception and communal attention, which may also help with heritage-related event detection and contribute to further planning and management strategies [9,42]. Data bundling and aggregation in the social domain, on the other hand, could help to profile the interests of user communities or user groups (e.g., local residents and tourists), which is beneficial for instance in devising recommendation systems. As argued in Section 2.6, multiple posts by the same user were not necessarily considered redundant in this study. Instead, the consistency and/or variations revealed in posted content by the same user [community/group] profile could further categorize their preference and opinions related to the cultural significance of heritage [117].

### 4.2. Limitations and Future Steps

No thorough human evaluations and annotations were performed during the construction of the datasets presented in this paper. This manuscript provides a way to circumvent this step by using only the confidence and [dis-]agreement of presumably well-trained models as a proxy for the more conventional "inter-annotator" agreement to show the quality of datasets and generate [pseudo-]labels [46]. This resembles the idea of using the consistency, confidence, and disagreement to improve the model performance in semi-supervised learning [48,49,91]. For the purpose of introducing a general framework that could generate more graph datasets, it is preferable to exclude humans from the loop as this would function as a bottleneck limiting the process, both in time and monetary resources, and in demanded domain knowledge. However, for applications where more accurate conclusions are needed, human evaluations on the validity, reliability, and coherence of the models are still needed. In order to gain a clear sense of the performance before implementation, the inspection of some predicted results is a prudent suggestion. As the step of [pseudo-]label generation was relatively independent from the other steps introduced in

this paper, higher-quality labels annotated and evaluated by experts and/or crowd-workers could still be added at a later stage as augmentation or even replacement, as an active learning process [45,47,89]. For example, future studies are invited to integrate the more recognized classification frameworks for heritage values and heritage attributes [3–5], in response to the possible imprecision of concepts as pointed out in Section 2.5. Moreover, generating labels of heritage values and attributes was only a choice motivated by the use-case at hand which suffices to show the utility of the framework for exploratory analyses on attributed graphs in cases where the sources of data are inherently unstructured and the connections between data points are inherently multi-faceted. Yet, it is also possible to apply the same framework as well as parts of the implemented workflow while only replacing the classifiers mentioned in Section 2.5 with domain-specific modules appropriated to the use-cases, to answer other exploratory questions in urban data science and computational social sciences, as suggested in Section 2.1.

While scaling up the dataset construction process, such as from VEN to VEN-XL, a few changes need to be adopted. For data collection, an updated strategy is described in Appendix A. For feature and label generation, mini-batches and GPU computing significantly accelerated the process. However, the small graphs from case study cities containing around 3000 nodes already contained edges at the scale of millions, making it challenging to scale up in cases such as VEN-XL, the adjacency list of which would be at the scale of billions, easily exceeding the limits of computer memory. As a result, VEN-XL has not yet been constructed as a multi-graph. Further strategies such as using sparse matrices [123] and parallel computing should be considered. Moreover, the issue of scalability should also be considered for later graph neural network training, since the multi-graphs constructed in this study can become quite dense locally. Sub-graph sampling methods should be applied to avoid "neighbourhood explosion" [44].

Although the motivation of constructing datasets regarding heritage values and attributes from social media was to promote inclusive planning processes, the selection of social media platforms already automatically excluded those not using, or not even aware of, the platform, let alone those not using internet. The scarce usage of Flickr in China, as an example of its limitation, also suggested that conclusions drawn from such datasets may reflect perspectives from the "tourist gaze" [124] rather than local communities, and therefore losing some representativeness and generality. However, the main purpose of this paper is to provide a reproducible methodological framework with mathematical definitions, not limited to Flickr as a specific instance. Images, texts, and even audio files and videos from other platforms such as Weibo, Dianping, RED, and TikTok that are more popular in China could also add complementary local perspectives. With careful adaptions, archives, official documents, news articles, academic publications, and interview transcripts could also be constructed in similar formats for fairer comparisons, which again would fit in the general framework proposed in Section 2.1 as specified instances.

## 5. Conclusions

This paper introduced a novel methodological framework to construct graph-based multi-modal datasets *Heri-Graphs* concerning heritage values and attributes using data from the social media platform Flickr. Pre-trained machine learning models were applied to generate multi-modal features and domain-specific pseudo-labels. A full mathematical formulation is provided for the feature extraction, label generation, and graph construction processes. Temporal, spatial, and social relationships among the posts are used to construct multi-graphs, ready to be utilised as the contextual information for further semi-supervised machine learning tasks. Three case study cities, namely Amsterdam, Suzhou, and Venice containing UNESCO World Heritage properties are tested with the framework to construct sample datasets, being evaluated and filtered with the consistency of models and qualitative inspections. The datasets in the three sample cities are shown to provide meaningful information concerning the spatio-temporal and socio-economic distributions of heritage values and attributes conveyed by social media users, useful for knowledge documentation

and mapping for heritage and urban studies. Such understanding is strongly aligned with the Sustainable Development Goal (SDG) 11, with its ultimate objective of making the urban heritage management processes more inclusive. The datasets created through the proposed framework provide a basis for revisiting or generalizing the *First Law of Geography* as formulated by Tobler to include the new senses of proximity or similarity caused by crowd behaviour and other social connections through electronic media that are arguably not directly related to geographical matters. This is especially important since heritage studies in particular, urban studies, and computational social sciences in general are almost always concerned with contextual information, which is arguably not limited to the geographical context but also to the social and temporal contexts. Such datasets have the potential to be applied by both the machine learning community and urban data scientists to help answer interesting questions of scientific/technical and social relevance, which could also be applied globally with a broad range of use cases.

**Author Contributions:** Conceptualization, Nan Bai, Pirouz Nourian, Renqian Luo and Ana Pereira Roders; Methodology, Nan Bai and Renqian Luo; Software, Nan Bai; Validation, Nan Bai, Pirouz Nourian, Renqian Luo and Ana Pereira Roders; Formal Analysis, Nan Bai and Pirouz Nourian; Investigation, Nan Bai; Resources, Pirouz Nourian and Ana Pereira Roders; Data Curation, Nan Bai; Writing—Original Draft Preparation, Nan Bai; Writing—Review and Editing, Nan Bai, Pirouz Nourian, Renqian Luo and Ana Pereira Roders; Visualization, Nan Bai and Pirouz Nourian; Supervision, Pirouz Nourian and Ana Pereira Roders; Project Administration, Ana Pereira Roders; Funding Acquisition, Ana Pereira Roders. All authors have read and agreed to the published version of the manuscript.

**Funding:** The presented study is within the framework of the Heriland-Consortium. HERILAND is funded by the European Union's Horizon 2020 research and innovation programme under the Marie Sklodowska-Curie grant agreement No. 813883.

**Institutional Review Board Statement:** The study was approved by the Institutional Review Board of TU Delft with Data Protection Impact Assessment (DPIA).

**Informed Consent Statement:** Not applicable.

**Data Availability Statement:** The dataset and codes to generate datasets introduced in this paper can be found at https://github.com/zzbn12345/Heri_Graphs (accessed on 31 May 2022).

**Acknowledgments:** This project collected data from the Flickr social media platform only when the original owners had indicated that their posted images were "downloadable". The copyright of all the downloaded and processed images belongs to the image owners. We are grateful to the constructive and insightful suggestions and comments from the anonymous reviewers and the editors, the discussions from our fellow colleagues, and the support from the Heriland community.

**Conflicts of Interest:** The authors declare no conflict of interest. The funders had no role in the design of the study; in the collection, analyses, or interpretation of data; in the writing of the manuscript, or in the decision to publish the results.

## Abbreviations

The following abbreviations are used in this manuscript:

| | |
|---|---|
| Acc | Accuracy |
| AMS | Data of Amsterdam, The Netherlands |
| API | Application Programming Interface |
| BC-SVM | Bagging Classifier with SVMs as the internal base estimators |
| BERT | Bidirectional Encoder Representations from Transformers |
| CNN | Convolutional Neural Network |
| CC | Connected Components |
| CV | Cross-Validation |
| GNB | Gaussian Naive Bayes |
| HA | Heritage Attributes |

| HUL | Historic Urban Landscape |
| HV | Heritage Values |
| KNN | K-Nearest Neighbours |
| ML | Machine Learning |
| MLP | Multi-layer Perceptron |
| MML | Multi-modal Machine Learning |
| NLP | Natural Language Processing |
| OUV | Outstanding Universal Value |
| RF | Random Forest |
| SDG | Sustainable Development Goal |
| SNA | Social Network Analysis |
| SUZ | Data of Suzhou, China |
| SVM | Support Vector Machine |
| UGC | User-Generated Content |
| ULMFiT | Universal Language Model Fine-tuning |
| UNESCO | The United Nations Educational, Scientific and Cultural Organization |
| VEN | Data of Venice, Italy |
| VEN-XL | The extra-large version of Venice data |
| w. | with |
| WH | World Heritage |
| WHL | World Heritage List |

## Appendix A. Details of Collecting the Raw Dataset

For each case study city, FlickrAPI python library was used to access the `photo.search` API method provided by Flickr, using the Geo-locations in Table 1 as the centroids to search the IDs of geo-tagged images within a fixed radius (5 km for Venice and Suzhou, and 2 km for Amsterdam) that covers the major urban area of the corresponding UNESCO World Heritage property. To construct comparable and compatible datasets from the three cities, a maximum of 5000 IDs were given to the search engine for each city, since Flickr users are relatively scarce in China. For all the image IDs, only those with a flag of `candownload` indicated by the owner were further queried, in order to respect the privacy and copyrights of Flickr users. Those images were further queried through `photo.getInfo` and `photo.getSizes` API methods to retrieve the following information: the owner's ID; the owner's registered location on Flickr; the title, description, and tags provided by the user; the geo-tag of the image; the timestamp of the image marking when it was taken, and URLs to download the `Large Square` (150 × 150 px) and `Small 320` (320 × 240 px) versions of the original image. The images that contained the user tag of "erotic" were excluded from the query. Then, all the images of both sizes were saved and transformed into RGB format as raw visual data.

A stop-word list was used to remove the HTML symbols and other formatting elements from the texts and to filter out textual data that were mainly (1) a description of the camera used, (2) a default image name generated by the camera, (3) an advertisement or a promotion. Google Translator API from the Deep Translator python library was used to detect the languages of the posts on the sentence level to mark whether the sentence was written in English, the local language (Dutch, Chinese, and Italian, respectively, for the three cities), or any other languages. The same API was used to translate the non-English sentences into English. Then, all the *valid* sentences from any textual field of the same post were merged into a new field named `Revised Text` as the raw textual data.

Furthermore, the public contact lists and group (subscription) lists of all the retrieved owners were queried through the `contacts.getPublicList` and the `people.getPublicGroups` API methods, while all user and group information was only considered as a [semi]-anonymous ID with respect to the privacy policy.

To test the scalability of the methodological workflow, another larger dataset without the limit of maximum 5000 IDs was also collected in Venice (VEN-XL). The API of Flickr has a limitation at the scale of queries, which would return occasional errors during times when the server was experiencing functional issues. This requires a different strategy during data

collection of the larger dataset. In this study, a $20 \times 20$ grid was tiled in the area of Venice (from 45.420855′ N 12.291054′ E to 45.448286′ N 12.369234′ E) to collect the post IDs from the centroid of each tile within a radius of 0.3 km, which were later aggregated by removing the duplicated IDs collected by multiple tiles to form the entire large dataset, similar to the strategy applied by Bekker [125]. The further steps of data cleaning and pre-processing remained the same with the smaller datasets.

## Appendix B. Details for Machine Learning Models

A dataset with 902 sample images collected in Tripoli, Lebanon and classified with expert-based annotations presented in Ginzarly et al. [26] was used to train several ML models to replicate the experts' behaviour on classifying depicted scenery. For each image, a unique class label among the 9 depicted scenes mentioned in Table A4 was provided. In total, 10% of the images were separated and kept away during training as the test dataset and the remaining 812 images were used to train ML models with Scikit-learn python library [95]. Among the 812 data points, `train_test_split` method of the library was further used to split out a validation dataset with 203 samples (25%). The 512-dimensional visual representation introduced in Section 2.4.1 was generated from the images as the input of ML models, while the class label was used as categorical output of the multi-class single-label classification task.

For each of the selected ML models, `GridSearchCV` function with 10-fold cross-validation was used to wrap the model with a set of tunable parameters in a small range to be selected, while the average top-1 accuracy was used as the criterion for model selection. All 812 samples were input to the cross-validation (CV) to tune the hyper-parameters, after which the trained models with their optimal hyper-parameters were tested on the 203 validation data samples and the unseen test dataset with the remaining 90 samples. For the latter steps, the top-1 accuracy and macro-average F1 scores (harmonic average of the precision and recall scores) of all classes were used as the evaluation metrics. All experiments were conducted using a 12th Gen Intel(R) Core(TM) i7-12700KF CPU.

The implementation details of the models are as follows:

### Appendix B.1. MLP

The model used L2 penalty of $1 \times 10^{-4}$, solver of stochastic gradient descent, adaptive learning rate and early stopping with maximum 300 iterations. It was tuned on the initial learning rate in $\{0.05, 0.1, 0.2\}$, and hidden sizes of one layer in $\{32, 64, 128, 256\}$ or two layers in $\{(256, 128), (256, 64), (256, 32), (128, 64), (128, 32)\}$. The best model had two hidden layers of (256,128) with a learning rate of 0.05.

### Appendix B.2. KNN

The model was tuned on the number of neighbours in range $[3, 11] \subset \mathbb{N}$, and the weights of uniform, Manhattan distance, or Euclidean distance. The best model had six neighbours in Euclidean distance.

### Appendix B.3. GNB

The model did not have a tunable hyper-parameter.

### Appendix B.4. SVM

The model was tuned on the kernel type in {linear, poly, rbf, sigmoid}, regularization parameter *C* in range $[0.1, 2.0] \subset \mathbb{R}$, kernel coefficient gamma in {scale, auto}, and degree of the polynomial kernel function in range $[2, 4] \subset \mathbb{N}$. The best model used RBF kernel with scaled weights and regularization parameter of 1.8.

*Appendix B.5. RF*

The model did not restrict the maximum depth of the trees. It was tuned on the class weight in settings of uniform, balanced, and balanced over sub-samples, and the minimum samples required to split a tree node in $\{2, 7, 12, \dots, 97\}$. The best model had a balanced class weight and a minimum of 17 samples to split a tree node.

*Appendix B.6. Bagging*

The model had 10 base estimators in the ensemble. It was tuned on the base estimator in SVM, Decision Tree, and KNN classifiers, and the proportion of maximum features used to train internal weak classifiers within the range $[0.1, 1.0] \subset \mathbb{R}$. The best model used maximum 50% of all features to fit SVM as internal base estimator.

*Appendix B.7. Voting*

The model took the first six aforementioned trained models as inputs in the ensemble to vote for the output and was tuned on the choice of hard (voting on top-1 prediction) and soft (voting on the averaged logits) voting mechanism. The best model used the soft voting mechanism.

*Appendix B.8. Stacking*

The model stacked the outputs of the first six aforementioned trained models in the ensemble followed by a final estimator and was tuned on the choice of final estimator among SVM and Logistic Regression. The best model used Logistic Regression as the final estimator.

**Appendix C. Nomenclature**

Tables A1 and A2 give an overview of the mathematical notations and functions used in this paper.

**Table A1.** The nomenclature of mathematical notations used in this paper in alphabetic order. All superscripts of matrices are merely tags, not to be confused with exponents and operations, with the exception of transpose operator $\square^{\mathsf{T}}$.

| Symbol | Data Type/Shape | Description |
|---|---|---|
| $A$ | Matrix of Boolean $A := (A^{\text{TEM}} > 0) \bigvee (A^{\text{SOC}} > 0) \bigvee (A^{\text{SPA}} > 0) \in \{0,1\}^{K \times K}$ | The adjacency matrix of all post nodes in the set $\mathcal{V}$ that have at least one link connecting them as a composed simple graph. |
| $A^{(*)}$ | Matrix of Float $A^{(*)} := [a_{i,i'}^{(*)}]_{K \times K} \in \mathbb{R}^{K \times K}$, $A^{(*)} \in \{A^{\text{TEM}}, A^{\text{SOC}}, A^{\text{SPA}}\}$ | The weighted adjacency matrix of each of the three sub-graphs $\mathcal{G}^{(*)}$ of the multi-graph $\mathcal{G}$, "(*)" represents one of the link types in {TEM, SOC, SPA}. |
| $A^{\mathcal{U}}$ | Matrix of Boolean $A^{\mathcal{U}} := [a_{j,j'}^{\mathcal{U}}]_{|\mathcal{U}| \times |\mathcal{U}|} \in \{0,1\}^{|\mathcal{U}| \times |\mathcal{U}|}$ | The adjacency matrix of all unique users $\mathcal{U}$ marking their direct friendship which also included the relationship among themselves. |
| $A^{\mathcal{U}'}$ | Matrix of Float $A^{\mathcal{U}'} := [a_{j,j'}^{\mathcal{U}'}]_{|\mathcal{U}| \times |\mathcal{U}|} \in [0,1]^{|\mathcal{U}| \times |\mathcal{U}|}$ | The weighted adjacency matrix of all unique users $\mathcal{U}$ marking their mutual interest in terms of the Jaccard Index of the public groups that they follow. |
| $\alpha_{\mathcal{T}}, \alpha_{\mathcal{U}}^{(n)}$ | Float scalars $\alpha_{\mathcal{T}}, \alpha_{\mathcal{U}}^{(1)}, \alpha_{\mathcal{U}}^{(2)}, \alpha_{\mathcal{U}}^{(3)} \in [0,1]$ | Parameters adjusting the weights of linear combination in relationship matrices $\mathfrak{T}$ and $\mathfrak{U}$. |
| $\beta_{\mathcal{U}}$ | Float scalar $\beta_{\mathcal{U}} \in (0,1)$ | The threshold to define mutual interest of two users as the Jaccard Index of public groups. |
| $\chi^2$ | Float Scalar | The Chi-square statistics of two distributions. |

**Table A1.** *Cont.*

| Symbol | Data Type/Shape | Description |
|---|---|---|
| $\mathfrak{d}_i, \mathfrak{D}$ | Object Tuples $\mathfrak{d}_i = (\mathfrak{I}_i, \mathcal{S}_i, \mathfrak{u}_i, t_i, \mathfrak{l}_i), \mathfrak{d}_i \in \mathfrak{D} = \{\mathfrak{d}_1, \mathfrak{d}_2, \ldots, \mathfrak{d}_K\}$ | The tuple of all raw data (image, sentences, user ID, timestamp, and geo-location) from one sample point. |
| $D_{\text{KL}}$ | Float Scalar | The Kullback–Leibler (KL) divergence of two distributions. |
| $\varepsilon$ | Float Scalar | An arbitrary small number to avoid zero-division. |
| $F$ | Matrix of Integers and Floats $\boldsymbol{F} = [\mathfrak{f}_i]_{3 \times K}, \mathfrak{f}_i = [\mathfrak{f}_{1,i}, \mathfrak{f}_{2,i}, \mathfrak{f}_{3,i}], \mathfrak{f}_{1,i} \in \mathbb{N}, \mathfrak{f}_{2,i}, \mathfrak{f}_{3,i} \in [0,1]$ | The face recognition result of an image sample in terms of the number of faces detected $\mathfrak{f}_{1,i}$, the model confidence for the prediction $\mathfrak{f}_{2,i}$, and the proportion of total area of bounding boxes of detected faces to the total area of images $\mathfrak{f}_{3,i}$. |
| $G_0$ | Undirected weighted graph $G_0 = (V_0, E_0, \boldsymbol{w}_0)$ | The complete spatial network in a city weighted by the travel time with all sorts of transportation between spatial nodes. |
| $G$ | Undirected weighted graph $G = (V, E, \boldsymbol{w}), V \subseteq V_0, E \subseteq E_0, \boldsymbol{w} \subseteq \boldsymbol{w}_0$ | The spatial network in a city weighted by the travel time between spatial nodes (no more than 20 min) that have at least one sample posted near them. |
| $\mathcal{G}$ | Weighted multi-graph $\mathcal{G} = (\mathcal{V}, \{\mathcal{E}^{\text{TEM}}, \mathcal{E}^{\text{SOC}}, \mathcal{E}^{\text{SPA}}\}, \{\boldsymbol{w}^{\text{TEM}}, \boldsymbol{w}^{\text{SOC}}, \boldsymbol{w}^{\text{SPA}}\})$ | The graph including the temporal, social, and spatial links $\mathcal{E}^{(*)}$ among the post nodes from set $\mathcal{V}$, weighted by the respective connection strengths $\boldsymbol{w}^{(*)}$. |
| $\mathcal{G}^{(*)}$ | Undirected weighted graph $\mathcal{G}^{(*)} = (\mathcal{V}, \mathcal{E}^{(*)}, \boldsymbol{w}^{(*)}), \mathcal{G}^{(*)} \in \{\mathcal{G}^{\text{TEM}}, \mathcal{G}^{\text{SOC}}, \mathcal{G}^{\text{SPA}}\}$ | The sub-graph of the multi-graph $\mathcal{G}$, while "(*)" represents one of the link types in {TEM, SOC, SPA}. |
| $H^{\text{B}}$ | Matrix of Floats $\boldsymbol{H}^{\text{B}} = [h_i^{\text{BERT}}]_{768 \times K}$ | The last hidden layer for `[CLS]` token of BERT model pre-trained on *WHOSe_Heritage*. |
| $H^{\text{v}}$ | Matrix of Floats $\boldsymbol{H}^{\text{v}} = [h_i^{\text{v}}]_{512 \times K}$ | The last hidden layer of ResNet-18 model pre-trained on *Places365*. |
| $i, i'$ | Integer Indices $i, i' \in \{1, 2, .., K\} \subset \mathbb{N}$ | The index of samples in the dataset $\mathfrak{D}$ of one case city. |
| $\mathfrak{I}_i$ | Tensor of Integers within $[0, 255] \in \mathbb{N}$ of size $150 \times 150 \times 3$ or $320 \times 240 \times 3$ | The raw image data of one sample post with RGB channels. |
| $I$ | Matrix of Boolean $\boldsymbol{I} \in \{0, 1\}^{|\mathcal{U}| \times |\mathcal{U}|}$ | The diagonal identity matrix marking the identity of unique users in $\mathcal{U}$. |
| $j, j'$ | Integer Indices $j, j' \in \{1, 2, .., |\mathcal{U}|\} \subset \mathbb{N}$ | The index of users in the ordered set $\mathcal{U}$ of all unique users from one case city. |
| $k$ | Integer Indices $k \in \{1, 2, .., |\mathcal{T}|\} \subset \mathbb{N}$ | The index of timestamps in the ordered set $\mathcal{T}$ of all unique timestamps from one case city. |
| $K$ | Integer $K = |\mathfrak{D}|$ | The sample size (number of posts) collected in one case city. |
| $K^{\text{HA}}$ | Matrix of Floats $\boldsymbol{K}^{\text{HA}} = [\kappa_i^{\text{HA}}]_{2 \times K}$ | The confidence indicator matrix for heritage attributes labels including the top-$n$ confidence and agreement between VOTE and STACK models. |
| $K^{\text{HV}}$ | Matrix of Floats $\boldsymbol{K}^{\text{HV}} = [\kappa_i^{\text{HV}}]_{2 \times K}$ | The confidence indicator matrix for heritage values labels including the top-$n$ confidence and agreement between BERT and ULMFiT models. |
| $l, l'$ | Integer Indices $l, l' \in \{1, 2, .., |V|\} \subset \mathbb{N}$ | The index of nodes in the ordered set $V$ of all spatial nodes from one case city. |
| $\mathfrak{l}_i$ | Tuple of Floats $\mathfrak{l}_i = (\mathfrak{x}_i, \mathfrak{y}_i)$ | The geographical coordinate of latitude ($\mathfrak{y}_i$) and longitude ($\mathfrak{x}_i$) as location of one sample. |
| $L^{\text{a}}$ | Matrix of logit vectors $\boldsymbol{L}^{\text{a}} = [l_i^{\text{a}}]_{102 \times K}$ | The last softmax layer of ResNet-18 model pre-trained on *SUN* predicting scene attributes. |
| $L^{\text{s}}$ | Matrix of logit vectors $\boldsymbol{L}^{\text{s}} = [l_i^{\text{s}}]_{365 \times K}$ | The last softmax layer of ResNet-18 model pre-trained on *Places365* predicting scene categories. |
| $\mathcal{M}$ | A set of objects | The set of machine learning models used to train classifiers on Tripoli data. |
| $O$ | Matrix of Boolean $\boldsymbol{O} := [\mathbf{o}_i] \in \{0, 1\}^{3 \times K}$ | The language detection result of the original language appearance of the sentences in each sample, in terms of English $\mathbf{o}_1$, local language $\mathbf{o}_2$, and other languages $\mathbf{o}_3$. |
| $R, R^{(*)}$ | Matrix of Float $\boldsymbol{R}, \boldsymbol{R}^{(*)} \in \mathbb{R}^{N \times K}, \boldsymbol{R}^{(*)} \in \{\boldsymbol{R}^{\text{TEM}}, \boldsymbol{R}^{\text{SOC}}, \boldsymbol{R}^{\text{SPA}}\}$ | The embedding matrices of each of the samples to a $N$-dimensional vector based on the general structure of the multi-graph $\mathcal{G}$ and the specific types of links. |
| $\mathcal{S}_i$ | Set of Strings $\mathcal{S}_i = \{\int_i^{(1)}, \int_i^{(2)}, \ldots, \int_i^{(|\mathcal{S}_i|)}\}$ or Empty Set $\mathcal{S}_i = \varnothing$ | The processed textual data as a set of individual sentences that have a valid semantic meaning and have been translated into English. |
| $S$ | Boolean Matrix $\boldsymbol{S} := [s_{l,i}] \in \{0, 1\}^{|V| \times K}$ | The one-hot embedding matrix of the samples corresponding to the geo-node set $V$. |
| $\mathfrak{S}$ | Matrix of Float $\mathfrak{S} := [\mathfrak{s}_{l,l'}] \in [0, 1]^{|V| \times |V|}$ | A matrix marking the spatial closeness of all the unique spatial nodes from set $V$ that can be reached within 20 min. |
| $\mathcal{T}$ | An ordered Set $\mathcal{T} = \{\tau_1, \tau_2, \ldots, \tau_{|\mathcal{T}|}\}$ | The ordered set of all unique timestamps from one case city. |

**Table A1.** *Cont.*

| Symbol | Data Type/Shape | Description |
|---|---|---|
| $\tau_k$ | Timestamp $\tau_k \in \mathcal{T}$ | A timestamp in the ordered set $\mathcal{T}$ of all unique timestamps. |
| $t_i$ | Timestamp $t_i \in \mathcal{T}$ | A timestamp indexed with sample ID in the ordered set $\mathcal{T}$ of all unique timestamps. |
| $T$ | Boolean Matrix $T := [t_{k,i}] \in \{0,1\}^{\lvert \mathcal{T} \rvert \times K}$ | The one-hot embedding matrix of the samples corresponding to the timestamp set $\mathcal{T}$. |
| $\mathfrak{T}$ | Matrix of Float $\mathfrak{T} \in [0,1]^{\lvert \mathcal{T} \rvert \times \lvert \mathcal{T} \rvert}$ | A matrix marking the temporal similarity of all the unique timestamps from set $\mathcal{T}$. |
| $\mathcal{U}$ | An ordered Set $\mathcal{U} = \{\mu_1, \mu_2, \ldots, \mu_{\lvert \mathcal{U} \rvert}\}$ | The ordered set of all unique users from one case study city. |
| $\mu_j$ | User ID Object $\mu_j \in \mathcal{U}$ | An instance of user in the ordered set $\mathcal{U}$ of all unique users. |
| $u_i$ | User ID Object $u_i \in \mathcal{U}$ | An instance of user indexed with sample ID in the ordered set $\mathcal{U}$ of all unique users. |
| $U$ | Boolean Matrix $U := [u_{j,i}] \in \{0,1\}^{\lvert \mathcal{U} \rvert \times K}$ | The one-hot embedding matrix of the samples corresponding to the user set $\mathcal{U}$. |
| $\mathfrak{U}$ | Matrix of Float $\mathfrak{U} \in [0,1]^{\lvert \mathcal{U} \rvert \times \lvert \mathcal{U} \rvert}$ | A matrix marking the social similarity of all the unique users from set $\mathcal{U}$, as a linear combination of identity matrix $I$ and adjacency matrices $A^{\mathcal{U}}, A^{\mathcal{U}'}$. |
| $V$ | A set of nodes $V = \{v_1, v_2, \ldots, v_{\lvert V \rvert}\}$ | The set of all the spatial nodes that have at least one sample posted near them. |
| $v_l$ | Spatial node $v_l \in V$ | A node in the set $V$ of all spatial nodes that have at least one sample posted near them. |
| $\mathcal{V}$ | A set of nodes $\mathcal{V} = \{v_1, v_2, \ldots, v_K\}$ | The set of all nodes of posts in one case city. |
| $v_i$ | Post/Sample node $v_i \in \mathcal{V}$ | A node in the set $\mathcal{V}$ of all nodes of posts in one case city. |
| $w, w^{(*)}$ | Vector of Float $w := [w_e] \in [0,20]^{\lvert E \rvert}, w^{(*)} := [w_e^{(*)}] \in \mathbb{R}^{\lvert \mathcal{E} \rvert}, w^{(*)} \in \{w^{\text{TEM}}, w^{\text{SOC}}, w^{\text{SPA}}\}$ | The weight vector of spatial network $G$ and post graphs $\mathcal{G}^{\text{TEM}}, \mathcal{G}^{\text{SOC}}, \mathcal{G}^{\text{SPA}}$, these weights are directly interchangeable with the adjacency matrices. |
| $X^{\text{vis}}$ | Matrix of Floats and Integers $X^{\text{vis}}_{982 \times K} = \left[ H^{v\top}, F^{\top}, \sigma^{(5)}(L^s)^{\top}, \sigma^{(10)}(L^a)^{\top} \right]^{\top}$ | The final visual feature concatenating the hidden layer $H^v$, the face detection results $F$, the filtered top-5 scene prediction $\sigma^{(5)}(L^s)$, and the filtered top-10 attribute prediction $\sigma^{(10)}(L^a)$. |
| $X^{\text{tex}}$ | Matrix of Floats and Integers $X^{\text{tex}}_{771 \times K} = \left[ H^{B\top}, O^{\top} \right]^{\top}$ | The final textual feature concatenating the hidden layer $H^B$ of BERT on [CLS] token, and the original language detection results $O$. |
| $Y^{\text{HA}}$ | Matrix of Floats $Y^{\text{HA}} = [y_i^{\text{HA}}]_{9 \times K}$ | The final generated label of heritage attributes on 9 depicted scenes, as the average of prediction from VOTE and STACK models. |
| $Y^{\text{HV}}$ | Matrix of Floats $Y^{\text{HV}} = [y_i^{\text{HV}}]_{11 \times K}$ | The final generated label of heritage values on 10 OUV selection criteria and an additional negative class, as the average of prediction from BERT and ULMFiT models. |

**Table A2.** The nomenclature of functions defined and used in this paper in alphabetic order.

| Symbol | Data Type/Shape | Description |
|---|---|---|
| $\text{argmx}(l, n)$ | Function outputting a set of floats or objects | The set of largest $n$ elements of any float vector $l$. |
| $f_{\text{BERT}}(\mathcal{S} \vert \Theta_{\text{BERT}})$ | Function inputting a sentence/paragraph or a batch of sentences/paragraphs, outputting a vector or a matrix of vectors | The pre-trained uncased BERT model fine-tuned on *WHOSe_Heritage* with the model parameters $\Theta_{\text{BERT}}$ that can process some textual inputs into the 768-dimensional hidden output vector $h^{\text{BERT}}$ of the [CLS] token. |
| $f_{\text{ResNet-18}}(\mathfrak{I} \vert \Theta_{\text{ResNet-18}})$ | Function inputting a tensor or a batch of tensors, outputting three vectors or three matrices of vectors | The ResNet-18 model pre-trained on *Places365* dataset with the model parameters $\Theta_{\text{ResNet-18}}$ that can process the image tensor $\mathfrak{I}$ into the predicted vectors of scenes $l^s$, predicted vectors of attributes $l^a$, and the last hidden layer $h^v$. |
| $g_{\text{BERT}}(\mathcal{S} \vert \Theta_{\text{BERT}})$ | Function inputting a sentence/paragraph or a batch of sentences/paragraphs, outputting a vector or a matrix of vectors | The end-to-end pre-trained uncased BERT model fine-tuned on *WHOSe_Heritage* with the model parameters $\Theta_{\text{BERT}}$ together with the MLP classifiers that can process some textual inputs into the logit prediction vector $y^{\text{BERT}}$ of 11 heritage value classes concerning OUV. |

**Table A2.** *Cont.*

| Symbol | Data Type/Shape | Description |
|---|---|---|
| $g_{\mathrm{ULMFiT}}(\mathcal{S}\|\Theta_{\mathrm{ULMFiT}})$ | Function inputting a sentence/paragraph or a batch of sentences/paragraphs, outputting a vector or a matrix of vectors | The end-to-end pre-trained ULMFiT model fine-tuned on *WHOSe_Heritage* with the model parameters $\Theta_{\mathrm{ULMFiT}}$ together with the MLP classifiers that can process some textual inputs into the logit prediction vector $y^{\mathrm{ULMFiT}}$ of 11 heritage value classes concerning OUV. |
| $h_{\mathrm{VOTE}}(h^{\mathrm{v}}\|\Theta_{\mathrm{VOTE}},\mathcal{M},\Theta_{\mathcal{M}})$ | Function inputting a vector or a batch of vectors, outputting a vector or a matrix of vectors | The ensemble Voting Classifier with model parameter $\Theta_{\mathrm{VOTE}}$ of machine learning models from $\mathcal{M}$ with their respective model parameters $\Theta_{\mathcal{M}}$, which processes the visual feature vector $h^{\mathrm{v}}$ into the logit prediction vector $y^{\mathrm{VOTE}}$ of 9 heritage attribute classes concerning depicted scenes. |
| $h_{\mathrm{STACK}}(h^{\mathrm{v}}\|\Theta_{\mathrm{STACK}},\mathcal{M},\Theta_{\mathcal{M}})$ | Function inputting a vector or a batch of vectors, outputting a vector or a matrix of vectors | The ensemble Stacking Classifier with model parameter $\Theta_{\mathrm{STACK}}$ of machine learning models from $\mathcal{M}$ with their respective model parameters $\Theta_{\mathcal{M}}$, which processes the visual feature vector $h^{\mathrm{v}}$ into the logit prediction vector $y^{\mathrm{STACK}}$ of 9 heritage attribute classes concerning depicted scenes. |
| $\mathcal{I}(\mu_j)$ | Function outputting an ordered set of objects | The set of public groups that are followed by user $\mu_j$. |
| $\mathrm{IoU}(\mathcal{A},\mathcal{B})$ | Function outputting a non-negative float | The Jaccard Index of any two sets $\mathcal{A},\mathcal{B}$ as the cardinality of the intersection of the two sets over that of the union of them. |
| $\max(l,n)$ | Function outputting a float | The $n_{\mathrm{th}}$ largest element of any float vector $l$. |
| $\sigma^{(n)}(l)$ | Function both inputting and outputting a logit vector | The activation filter to keep the top-$n$ entries of any logit vector $l$ and smooth all the others entries based on the total confidence (sum) of top-$n$ entries. |

## Appendix D. Definition of Categories for Heritage Values and Attributes

Tables A3 and A4, respectively, give a detailed definition of heritage values (in terms of Outstanding Universal Value selection criteria) and heritage attributes (in terms of depicted scenes) categories applied in this paper.

**Table A3.** The definition for each UNESCO World Heritage OUV selection criterion as heritage value category in this dataset and its main topic according to previous scholars [2,54,55].

| Criterion | Focus | Definition |
|---|---|---|
| (i) | Masterpiece | *To represent a masterpiece of human creative genius;* |
| (ii) | Values/Influence | *To exhibit an important interchange of human values, over a span of time or within a cultural area of the world, on developments in architecture or technology, monumental arts, town-planning or landscape design;* |
| (iii) | Testimony | *To bear a unique or at least exceptional testimony to a cultural tradition or to a civilization which is living or which has disappeared;* |
| (iv) | Typology | *To be an outstanding example of a type of building, architectural or technological ensemble or landscape which illustrates (a) significant stage(s) in human history;* |
| (v) | Land-Use | *To be an outstanding example of a traditional human settlement, land-use, or sea-use which is representative of a culture (or cultures), or human interaction with the environment especially when it has become vulnerable under the impact of irreversible change;* |
| (vi) | Associations | *To be directly or tangibly associated with events or living traditions, with ideas, or with beliefs, with artistic and literary works of outstanding universal significance;* |
| (vii) | Natural Beauty | *To contain superlative natural phenomena or areas of exceptional natural beauty and aesthetic importance;* |
| (viii) | Geological Process | *To be outstanding examples representing major stages of earth's history, including the record of life, significant on-going geological processes in the development of landforms, or significant geomorphic or physiographic features;* |
| (ix) | Ecological Process | *To be outstanding examples representing significant on-going ecological and biological processes in the evolution and development of terrestrial, fresh water, coastal and marine ecosystems and communities of plants and animals;* |
| (x) | Bio-diversity | *To contain the most important and significant natural habitats for in situ conservation of biological diversity, including those containing threatened species of outstanding universal value from the point of view of science or conservation.* |

**Table A4.** The definition for depicted scenery as heritage attribute category in this dataset and its tangible/intangible type according to previous scholars [5,26,93].

| Attribute | Type | Definition |
|---|---|---|
| Monuments and Buildings | Tangible | *The exterior of a whole building, structure, construction, edifice, or remains that host(ed) human activities, storage, shelter or other purpose;* |
| Building Elements | Tangible | *Specific elements, details, or parts of a building, which can be constructive, constitutive, or decorative;* |
| Urban Form Elements | Tangible | *Elements, parts, components, or aspects of/in the urban landscape, which can be a construction, structure, or space, being constructive, constitutive, or decorative;* |
| Urban Scenery | Tangible | *A district, a group of buildings, or specific urban ensemble or configuration in a wider (urban) landscape or a specific combination of cultural and/or natural elements;* |
| Natural Features and Landscape Scenery | Tangible | *Specific flora and/or fauna, such as water elements of/in the historic urban landscape produced by nature, which can be natural and/or designed;* |
| Interior Scenery | Tangible/ Intangible | *The interior space, structure, construction, or decoration that host(ed) human activity, showing a specific (typical, common, special) use or function of an interior place or environment;* |
| People's Activity and Association | Intangible | *Human associations with a place, element, location, or environment, which can be shown with the activities therein;* |
| Gastronomy | Intangible | *The (local) food-related practices, traditions, knowledge, or customs of a community or group, which may be associated with a community or society and/or their cultural identity or diversity;* |
| Artifact Products | Intangible | *The (local) artifact-related practices, traditions, knowledge, or customs of a community or group, which may be associated with a community or society and/or their cultural identity or diversity.* |

## Appendix E. Multi-Graph Visualization

The connected components of each type of temporal, social, and spatial links in each case study city are visualized in Figure A1, respectively. The `spring_layout` algorithm of NetworkX python library with optimal distance between nodes `k` of 0.1 and random seed of `10396953` are used to output the graphs.

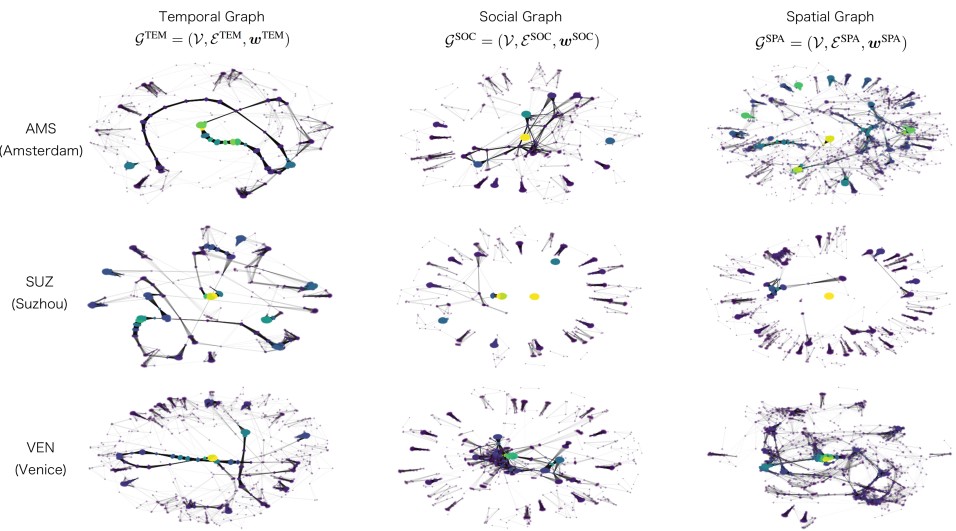

**Figure A1.** The subgraphs of the multi-graphs in each case study city visualized using spring layout in NetworkX. The node size and colour reflect the degrees, and link thickness the edge weights.

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
