# Peer review of "Heri-Graphs: A Dataset Creation Framework for Multi-Modal Machine Learning on Graphs of Heritage Values and Attributes with Social Media"

_ijgi, doi:10.3390/ijgi11090469_

Round 1

Reviewer 2 Report

Good for the publication in the present form

Author Response

We are grateful that the reviewer finds our paper good for publication in the present form.

Reviewer 3 Report

Thank you for the opportunity to review this manuscript. This study proposes a methodological workflow to derive information from UGC on social media resulting in multi-modal datasets for graph-based machine learning tasks concerning heritage values and attributes. The study is interesting, and the authors go to great lengths to describe the methods, including data processing, and presenting results.

I have the following comments:

In Section 2.2. Data collection and Pre-processing, the authors stated "FlickrAPI python library was used to access the API method provided by Flickr, using the Geo-locations in Table 1 as the centroids to search a maximum of 5000 IDs of geo-tagged images in a fixed radius covering the major urban area, to make the datasets from the three cities comparable and compatible." However, it is not completely clear why they chose the limit of 5000 geotagged images. In such cases, what are the implications of having a different number of users (e.g., 5000 images may correspond to 500 users in one area, or between 200-250 users in other area), in the construction of such datasets?;

Partly in keeping with the above comment one opportunity/recommendation is clearly the introduction of timestamps and other UGC features in datasets. What are the implications of building such datasets, using UGC that may correspond to distinct timeframes? Datasets may contain information created in a single year, or information that spans an extended period of time. What are the implications for generating visual and textual features, and for label generation process?

Finally, It would be valuable for the reader to further develop on the advantages and limitations of aggregating individual information (e.g., geographic coordinates and timestamps, visual and text information related to heritage values) using scale-dependent representations such as administrative levels, as well as on the related implications for constructing a collective sense of place. It might be of help Encalada et al. (2021) contribution;

References:

Encalada-Abarca L, Ferreira CC, Rocha J. (2021) Measuring Tourism Intensification in Urban Destinations: An Approach Based on Fractal Analysis. Journal of Travel Research. doi:10.1177/0047287520987627

Reviewer 4 Report

Thank you for the opportunity to contribute to this paper as a reviewer. All my opinion and recommendations have been made to contribute to this research paper.

The paper entitled “Heri-Graphs: A Workflow of Creating Datasets for Multi-modal Machine Learning on Graphs of Heritage Values and Attributes with Social Media” has the objective of proposing a workflow for creating a multi-modal dataset in the scope of Heritage with Social Media.

The paper presents the motivation, formal definitions and study cases as expected for a research paper. However, the current presentation could be improved in order to boost the expected paper contribution. The adjusts in presentation could be aggregated in these points:

* Insistent statement about state of the art: The authors mention several times “the usage of the state of the art” (e.g., CNN and BERT). However, it is expected that the authors took advantage of the state of the art, but for some particular instances of the proposed workflow, for example, different from Flickr, the state-of-the-art algorithm or method can be a different one. There are cases where GPT-2 is better than BERT.

* In my point of view the proposal is a framework and not just a workflow. From my perspective, when presenting technologies, instances, implementations and so on, the proposal goes further than a workflow definition. A workflow is expected to be a generic and abstract level of a pipeline, in which the input and output of each step are justified and organized to reach the final goal. Reading the paper, the idea of a framework sounds more tuned to the proposed idea.

* This is a very important point. It is not clear the workflow and the study case. The workflow (or framework) needs to be described following the expected contribution of each step. In the current manuscript, it is not clear the workflow instance for Flickr and the original (general) proposal. The title suggests the proposal as a general workflow, but the presentation and discussion are strictly related to Flickr and the technologies employed. For example, do I need to represent the visual information as a feature vector (i.e., using a CNN, Autoencoder etc) or do I need to use a Resnet-18 with Places365. The workflow (or framework) needs to be wide and instantiable to the most suitable technology, otherwise, your contribution will be stuck in your study case, without delivering an abstract model to be instantiated for a wide range of scenarios and their specificities. In this way, I’d like to strongly recommend a section devoted to explaining the proposal (input, processing, and output), and discussing the necessity of each process. After, in another section, the Flickr instance, justifies the selected technology inside the Flickr domain and its challenges.

Round 2

Reviewer 4 Report

Congratulations! This improved version contemplates all my questions.